# Comparative mucomic analysis of three functionally distinct *Cornu aspersum* Secretions

Antonio R. Cerullo[1,2,3], Maxwell B. McDermott[3,10], Lauren E. Pepi [4,10], Zhi-Lun Liu[1,5], Diariou Barry[1], Sheng Zhang[1], Xu Yang [4], Xi Chen [1,5,6,7], Parastoo Azadi[4], Mande Holford [2,3,6,8,9] & Adam B. Braunschweig [1,2,3,6] ✉

Every animal secretes mucus, placing them among the most diverse biological materials. Mucus hydrogels are complex mixtures of water, ions, carbohydrates, and proteins. Uncertainty surrounding their composition and how interactions between components contribute to mucus function complicates efforts to exploit their properties. There is substantial interest in commercializing mucus from the garden snail, *Cornu aspersum*, for skincare, drug delivery, tissue engineering, and composite materials. *C. aspersum* secretes three mucus—one shielding the animal from environmental threats, one adhesive mucus from the pedal surface of the foot, and another pedal mucus that is lubricating. It remains a mystery how compositional differences account for their substantially different properties. Here, we characterize mucus proteins, glycosylation, ion content, and mechanical properties that could be used to provide insight into structure-function relationships through an integrative "mucomics" approach. We identify macromolecular components of these hydrogels, including a previously unreported protein class termed Conserved Anterior Mollusk Proteins (CAMPs). Revealing differences between *C. aspersum* mucus shows how considering structure at all levels can inform the design of mucus-inspired materials.

Mollusca utilize mucus as glues[1–3], to create slick non-stick surfaces[4,5], and to facilitate innate immunity[6,7]. The metabolic costs of mucus production can exceed one-quarter of mollusks' energy budgets, indicating how important these materials are for survival[8]. The structural component differentiating mammalian mucus from other soft materials are mucins—proteins containing densely *O*-glycosylated repetitive regions that form crosslinked networks from disulfide bonds, ion-bridges, and carbohydrate binding[9,10]. However, molluscan mucus composition, and how they contribute to function, are not as well understood. Studies on *C. aspersum* mucus have focused on quantification of the protein within the mucus[11], bioactivity[12], the presence of antimicrobial peptides[13], or its ecological role[14]. While mucins

[1]The Advanced Science Research Center, Graduate Center of the City University of New York, 85 St. Nicholas Terrace, New York, NY 10031, USA. [2]The PhD Program in Biochemistry, Graduate Center of the City University of New York, 365 Fifth Avenue, New York, NY 10016, USA. [3]Department of Chemistry and Biochemistry, Hunter College, 695 Park Avenue, New York, NY 10065, USA. [4]Complex Carbohydrate Research Center, University of Georgia, 315 Riverbend Road, Athens, GA 30602, USA. [5]Department of Chemical Engineering, The City College of New York, New York, NY 10031, USA. [6]The PhD Program in Chemistry, Graduate Center of the City University of New York, 365 Fifth Avenue, New York, NY 10016, USA. [7]The PhD Program in Physics, Graduate Center of the City University of New York, 365 Fifth Avenue, New York, NY 10016, USA. [8]The PhD Program in Biology, Graduate Center of the City University of New York, 365 Fifth Avenue, New York, NY 10016, USA. [9]Department of Invertebrate Zoology, The American Museum of Natural History, New York, NY 10024, USA. [10]These authors contributed equally: Maxwell B. McDermott, Lauren E. Pepi. ✉e-mail: abraunschweig@gc.cuny.edu

have been identified in aquatic snails and other mollusks[15,16] that contain the canonical A–B–A structure generally associated with mucins, with cysteine-rich (A) domains at the head and tail for disulfide bridging, and serine(Ser)/threonine(Thr)-rich (B) domains in the center possessing abundant glycosylation, no such mucins have been identified in *C. aspersum*[9]. Rather, characterization of snail mucins has been limited to compositional analysis of amino acid and glycan residues, or studies on the molecular masses and hydrodynamic radii of the hydrogel particles[17]. These studies found that proteins in the mucus secretions of *C. aspersum* contain overabundances of Ser/Thr residues, *N*-acetylgalactosamine (GalNAc), galactose (Gal), and fucose (Fuc) glycans, and the proteins had average molecular masses of 30 kDa, while mammalian mucins are typically 100 kDa to 1 MDa with an abundance of sialic acids[18]. Broad proteomic analyses and profiling of snail mucus, focused on *C. aspersum* snail–snail signaling[19], microbial interactions[20], and comparison of proteins between multiple snail species[21]. These studies illustrate that molluscan mucus contain proteins and glycans that are not found in mammalian mucus. Researchers have also investigated the role of ions in snail mucus and correlated increased $CaCO_3$ content with increased mucus aggregation and adhesion[4,22,23]. Notably, these prior studies analyze crude mucus collections, rather than purified mucus samples that reflect the protein compositions of the gels, themselves.

Despite these efforts, it remains unclear how differences in protein structure, ion concentration, glycosylation, and other factors operate synergistically to account for the substantial diversity in mucus material properties[24]. Here, we apply a systematic comparative mucomic analysis—defined as the combination of genetic, chemical, and material studies to understand the structure-function relationships of mucus—of adhesive[23], lubricating[4], and protective[25] mucus isolated from *C. aspersum*ˌ which are named in accordance with the materials' ecological function (Fig. 1a). Transcriptomic and proteomic sequencing identified the proteins expressed in each mucus and their abundances. Glycomic mass spectrometry was then employed to identify the structures of the glycans decorating these proteins.

Elemental analysis through scanning electron microscopy (SEM) coupled with energy dispersive X-ray spectroscopy (EDX) measured concentrations of various ions in the materials. Atomic force spectroscopy quantified the mechanical properties (elastic modulus, $E$, and work of adhesion, $W$) of the three samples. Comparison of these datasets reveal how *C. aspersum* exploit differential protein expression—including a series of previously uncharacterized proteins—glycosylation, and ion concentration are used to explain how these hydrogels behave as adhesives, lubricants, or protective barriers[26] (Fig. 1b).

## Results

### Collection and purification of adhesive, lubricious, and protective snail mucus secretions

Adhesive, lubricious, and protective mucus samples were separately collected from *C. aspersum* snails (Supplementary Fig. 1)[27]. The snails were placed onto inverted petri dishes, to which they attached, resulting in the deposition of adhesive mucus from the pedal surface of the foot. Lubricating mucus was collected from the trails left behind by the pedal surfaces of snails that had crawled along petri dishes. Protective mucus was scraped from the dorsal surface of the snail. Proteins embedded within the mucus gel were fractionated from cellular detritus so that analysis focused upon the components integrated into the mucus (Supplementary Fig. 2). Isolated mucus all occurred as flocculent, beige substances with the consistency of cotton candy (Supplementary Fig. 3). Purified mucus proteins were resuspended and subjected to spectrophotometric analysis to quantify the yield of protein samples at every purification step (Supplementary Fig. 4, Supplementary Table 1). Initial protein concentrations in the solution of resuspended crude mucus were approximately 77–170 mg/mL for all three samples. Following purification, 3.9–7.4% of the initial total protein was recovered.

### Identification and sequence alignment of snail mucus proteins

Shotgun proteomic sequencing supported by a de novo assembled transcriptome identified proteins in the purified mucus samples[28]. As *C. aspersum*'s genome has not yet been sequenced, a transcriptomic reference database of actively translated genes found in mucus-producing tissue was produced from RNA extracted from the foot and back tissue of whole *C. aspersum* snails. It is important to note that shotgun proteomics only detects fragmented peptides and not the whole protein, and does not detect fragments that are present as a result of proteolytic cleavage[29]. From the 179,552 transcripts, 71 provided coding sequences for proteins based on the standard criteria of having a minimum of two identified peptides and a false discovery rate of less than 1.0%[19,30]. All proteins were quantified based on MS/MS proteomic abundance (Supplementary Figs. 5–7, Supplementary Table 2). Many of the proteins had sequence similarity (*E*-value < $10^{-4}$) to known proteins from snails or other mollusks, and were thus assigned identities corresponding to the specific proteins with which they shared highest similarity (Supplementary Table 3). More than 20% of the proteins in each mucus matched proteins from other snails that have no reported function, referred to herein as "Snail" proteins, or appeared to be completely unique, which are referred to as "Novel" proteins. To better understand the relationships between our sequenced proteins and those of other mollusks, and to determine the functions of the identified and unidentified proteins, dendrogram analysis was employed to cluster the genes by sequence similarity to each other and a set of NCBI reference proteins of similar families from mollusks (Fig. 2, Supplementary Fig. 8). These clusters were assigned broad categorizations as inhibitors, enzymes, mucins, matrix, network, lectins, or ion-binders. Notably, there were two mucin clades—one large cluster of mucins broken into three smaller clades, and one separate smaller clade. Using this approach, the 12 Snail proteins and the 6 Novel proteins were each

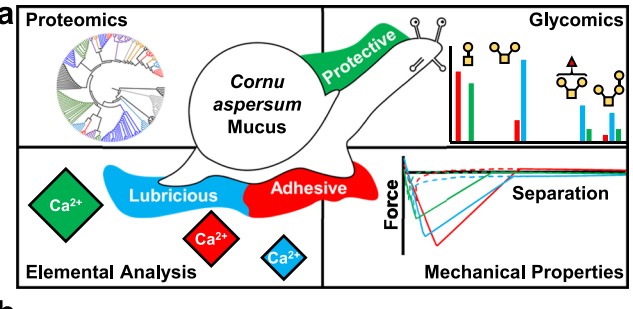

**Fig. 1 | Mucomic analysis of *Cornu aspersum* snail mucus composition.** **a** Adhesive, lubricating, and protective mucus are subjected to an omics-style analysis to understand the composition and properties. Red: adhesive; blue: lubricating; green: protective. **b** Comparative overview of the compositions and properties of *C. aspersum* adhesive, lubricious, and protective mucus.

|  | Adhesive | Lubricious | Protective |
|---|---|---|---|
| Collagens | Some | Many | Many |
| Ca-binders | None | Many | None |
| Crosslinkers | Many | Some | Some |
| Lectins | None | None | Many |
| Protease Inhibitors | Some | Some | Many |
| *O*-Glycans | Short | Long | Short |
| *N*-Glycans | Complex | Simple | Simple |
| Sialic Acids | Many | None | Some |
| $Ca^{2+}$ Content | ↑ | ↓ | ↑ |
| Young's Modulus | ↓ | ↑ | ↑ |
| Work of Adhesion | ↑ | ↓ | ↑ |

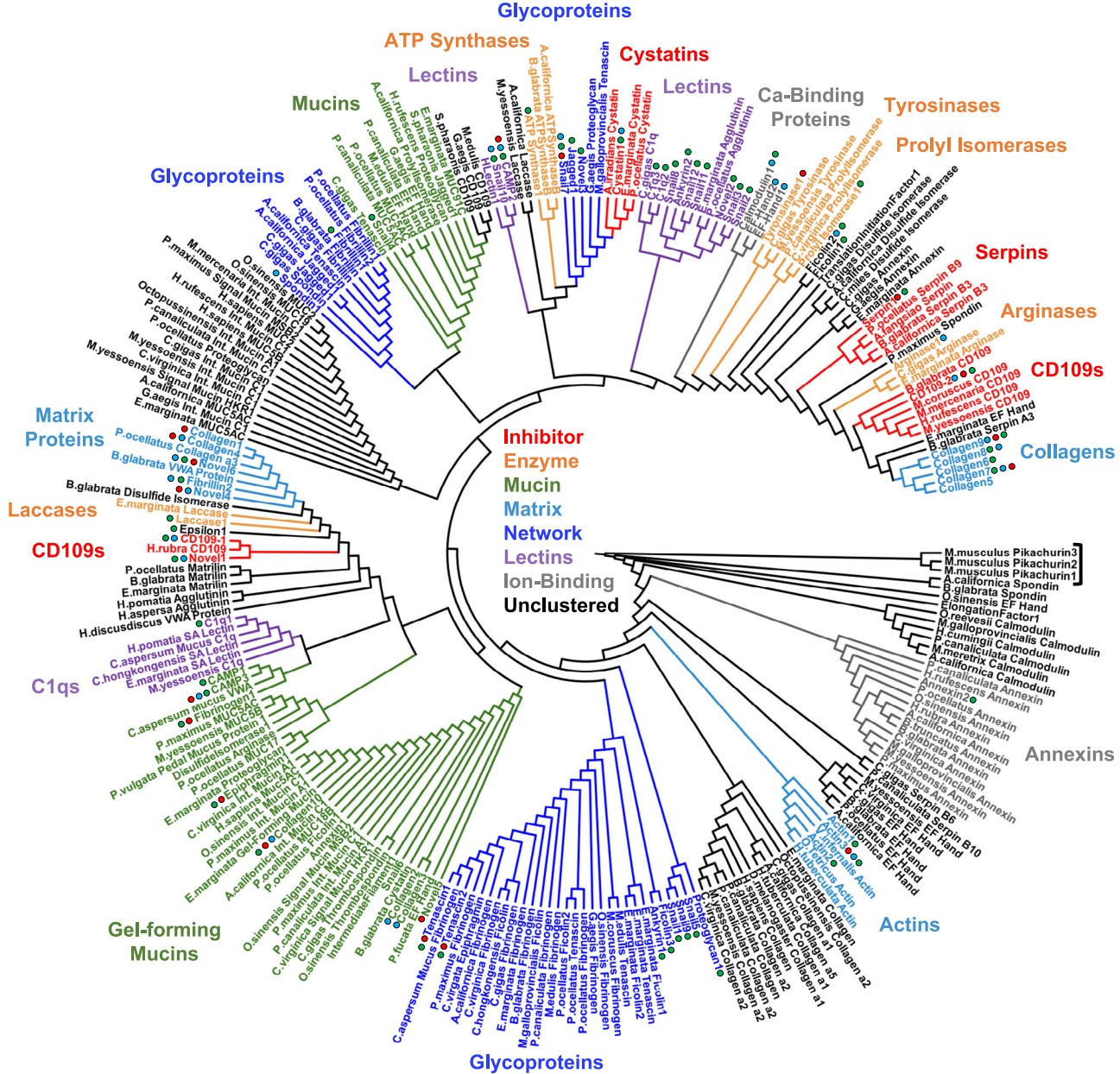

**Fig. 2 | Dendrogram of snail mucus proteins based on sequence similarity.** Clusters are colored according to protein function. The "Unclustered" (black) classification indicates a clade that had no discernible function or only contained reference proteins. Proteins identified in this study are labeled with circles. Circle color indicates the protein was found in adhesive (red), lubricating (blue), or protective (green) mucus. An outgroup, three *Mus musculus* proteins (Pikachurin1, Pikachurin2, Pikachurin3), is marked with a bracket. Dendrogram with branch lengths and included species is shown in Supplementary Fig. 8.

grouped into one of the aforementioned categories, allowing putative identifications that could not be made by BLASTP searches (Supplementary Table 4). An important limitation of the de novo transcriptomic approach is that the program SuperTranscripts produces the largest unspliced isoform of each protein sequence and the expressed proteins may have a lower molecular mass than predicted[31–33]. To further validate the molecular masses of the identified proteins, SDS-PAGE was conducted on each of the three mucus to resolve polypeptides by size. Lanes were divided into three molecular mass regions and excised for downstream proteomic analysis (Supplementary Figs. 9–11). 53 of the 71 assigned proteins were resolved by molecular mass and identified in this approach (Supplementary Figs. 12–15) and masses were consistent with those determined by transcriptomic analysis.

## Characterization of adhesive, lubricating, and protective *C. aspersum* mucus proteins

Several glycoproteins that are components of oligomeric networks (independent of conventional extracellular matrix proteins), such as fibrinogens, ficolins, and tenascins, were identified, suggesting a wide diversity occurs in the protein-protein networks. These proteins could, in turn, cause differences in the mechanical behavior of these mucus[34]. These glycoproteins make up about 15% of the adhesive and protective mucus, and 5% of the lubricating mucus, respectively. Importantly, BLASTP searches of several proteins returned A–B–A mucins as positive hits, which had not been identified in previous studies. It should be noted that mucins are challenging to identify via shotgun proteomics because their dense glycosylation limits enzymatic digestion[35], or via transcriptomics because of their tandem repeats[36]. A jagged-1-like

protein (Jagged1), which is involved in extracellular signaling pathways[37], had sequence similarity to MUC2 from *Pygocentrus nattereri* (Red-bellied piranha). A spondin-like protein (Spondin1), which mediates cell-extracellular matrix interactions[38], also displayed similarity to MUC2, MUC5AC, MUC12, MUC16, and MUC19 from mollusks and other marine life. Spondin1's sequence features several short regions that are either Ser- or Thr-rich. Curiously, these regions alternate between being Ser-rich and Thr-rich, meaning each region only incorporates one of these two amino acids. This protein is also 12% Cys by composition, which is more than five times greater than the natural abundance of Cys in invertebrate proteins[39], suggesting it has a propensity to form disulfide bridges. It is likely Spondin1 is a *C. aspersum* mucin because it contains repeating Ser/Thr-rich regions for potential *O*-glycosylation sites, similar to 'B' domains of mucins, and Cys-rich regions which can multimerize the protein by functioning like mucin 'A' domains. Vertebrate SCO-spondins, which are repetitive, highly glycosylated, and bind $Ca^{2+}$, and are known orthologs of invertebrate mucins[40]. From the alignment analysis, several proteins clustered within the large mucin families, including ones whose functions were not determined initially by BLAST. Snail6, Snail10, and Novel5 were clustered with mucins. Additionally, Snail1, Snail5, Snail9, and Novel3 fell within glycoprotein groupings. Thus, it appears that these secretions involve a combination of mucin-like proteins. Epiphragmin was identified in the adhesive mucus, which is used to create the epiphragm—the persistent glue that maintains bonds between the snail's shell and substrate[41]. This protein has been found previously in other snail species, such as the vineyard snail, *Cernuella virgata*, and is localized to the pedal surface of the foot[19], suggesting it is conserved in mollusks and has important function in snail adhesion. Notably, 30% of the adhesive mucus is composed of only two tenascin glycoproteins.

Extracellular matrix proteins comprise 40–50% of all three *C. aspersum* mucus protein samples, with lubricating mucus incorporating more matrix proteins (50%) than the other two (40% each). Eleven unique collagen genes were identified, which were found previously to be expressed in snail mucus[42]. While many of the matrix protein genes code for collagens, there are stark differences in abundances of these collagens between the samples. Collagen2, Collagen3, and Collagen11 are exclusive to lubricating mucus, while Collagen4, Collagen9, and Collagen10 are in all three mucus. Collagen7 and Collagen8 are more abundant in protective mucus than the other two. Collagen6 is exclusive to protective mucus. Collagen1 is found exclusively in adhesive mucus, albeit with very low abundance. Adhesive mucus shares Collagen4, Collagen7, Collagen9, and Collagen10 with the other mucus, however they are less abundant in adhesive mucus than the lubricating and protective[34].

Several enzymes that were found are involved in protein cross-linking, mucus network formation, or constructing biological glues, and likely serve a similar role in *C. aspersum* mucus[43,44]. Cysteines are abundant in mucins, and disulfide isomerases, like the identified DisulfideIsomerase1, construct mucus gels by catalyzing interchain disulfide bonds[44]. A prolyl isomerase, ProlylIsomerase1, was found, which has signaling and immune functions in mucus[45], and this class of proteins also regulates collagen crosslinking[46]. A tyrosinase, Tyrosinase1, was found exclusively in the adhesive mucus, which catalyzes the formation of L-DOPA from tyrosine. As L-DOPA is involved in forming strong glues in *Perna viridis* mussels[47], this observation suggests that *C. aspersum* land snails may use a similar adhesive mechanism as marine mollusks[47]. Tyrosinases also produce melanins, which are crosslinked networks formed through polymerization of phenolic molecules, and enzymes involved in melanin biosynthesis have been reported previously in snail and other invertebrate mucus secretions[48]. Additionally, a laccase, Laccase1, was identified, which catalyzes the oxidation and crosslinking of phenolic compounds[43]. Thus, tyrosinases, laccases, and other phenoloxidases may increase snail mucus integrity by crosslinking phenols of proteins and metabolites, similar to

mechanisms used in other mollusks[49]. No proteins strongly identifying with *P. viridis* mussel foot glue proteins were identified in this study, though Fibrillin1 and Fibrillin2 showed some sequence similarity to mussel foot proteins[47].

Several proteins were found that have potential roles in defense. All mucus include 1–5% protease inhibitor proteins, which can have infection-mitigating effects and also protect the protein scaffold from degradation[50]. Mucins and other mucus proteins are Ser- and Cys-rich, thus the serine (serpins and CD109s)[51,52] and cysteine (cystatins)[53] protease inhibitors identified in this analysis could prevent pathogens from degrading the mucus barrier. While serine protease inhibitors were found in all mucus, the adhesive mucus did not contain any cysteine protease inhibitors. Protective mucus is 10% lectins, which confer immune function in mollusks by protecting the snail's skin from pathogens[6]. The other two mucus were ~2% lectin. C1q lectins, which have immune function, complex with antigens, and noncovalently crosslink mucus glycoproteins[54], and Gal-specific H-type lectins[55], were exclusively found in the protective mucus. The protein C1q1 made up 6% of the protective secretion and <2% of the other two.

Calcium ion ($Ca^{2+}$)-binding proteins were entirely absent from the adhesive mucus and were minimally present in the protective mucus, but were abundant (~10%) in the lubricating mucus. This class of proteins includes annexins, calmodulins, and EF-hand proteins[56]. Since these proteins were mainly found in the lubricating mucus, they may play a different role outside of forming gel networks. These proteins are involved in $Ca^{2+}$-dependent signaling pathways, suggesting they relay environmental information back to mucus-producing tissue[57]. Prior research has demonstrated that $Ca^{2+}$ crosslinks mucus gel particles[4]. Thus, the presence of $Ca^{2+}$-binders suggests that $Ca^{2+}$ may have a different fate in the lubricating mucus than in adhesive and protective. It is possible these proteins function as ion traps, preventing the $Ca^{2+}$ from participating in ion bridges between highly glycosylated mucus proteins.

For each mucus, 20–40% of the identified proteins fell into the Snail and Novel groupings, meaning they had no identifiable function from BLASTP searches. Several of these proteins individually made up an appreciable amount of the mucus secretions. Over 40% of the adhesive mucus' is made of only 3 proteins, Snail7, Novel5, and Novel6. Novel5 comprised 7% of the adhesive mucus, where it was exclusively found. Snail7 and Novel6 were shared across all three mucus, but made up a much greater proportion of the adhesive sample than the other two. Using alignment analysis, all 18 of the Snail and Novel genes clustered into clades and assigned putative functions (Supplementary Table 3). Novel1 was grouped with CD109 proteins. Novel5 clustered with gel-forming mucins. Novel6 clustered within a family of mollusk glycoproteins and Von Willebrandt Factor A (VWA) proteins. Snail6, which was abundant primarily in lubricating mucus, was clustered into the same gel-forming mucin family. Snail4 was also placed with mucins. Snail7 fell within a clade of mollusk glycoproteins. Snail1, Snail5, Snail9, and Novel3 were clustered into mollusk glycoprotein clades. Snail2, Snail3, Snail8, Snail10, Snail11, Snail12, and Novel2 were grouped with lectins.

The SDS-PAGE analysis showed that all three mucus present a prominent glycoprotein band (Supplementary Figs. 9–11) at approximately 260 kDa, which was excised and subjected to proteomic analysis to identify the protein content (Supplementary Table 5). This analysis revealed that Collagen5, Collagen7, Novel4, a homolog of Novel4, which was named Novel4A, and two new proteins, deemed Novel7 and Novel8, which did not share similarity with any NCBI proteins, were found in these bands. Novel4, Novel7, and Novel8 were present in all samples. Further analysis of BLASTP results indicated Collagen5 and Collagen7 had similarities to oligomeric network-forming proteins[58]. PFAM domain search showed Collagen5 and Collagen7 contained von Willebrandt Factor A domains, calcium-binding regions, glycoprotein domains, disulfide-forming domains, and

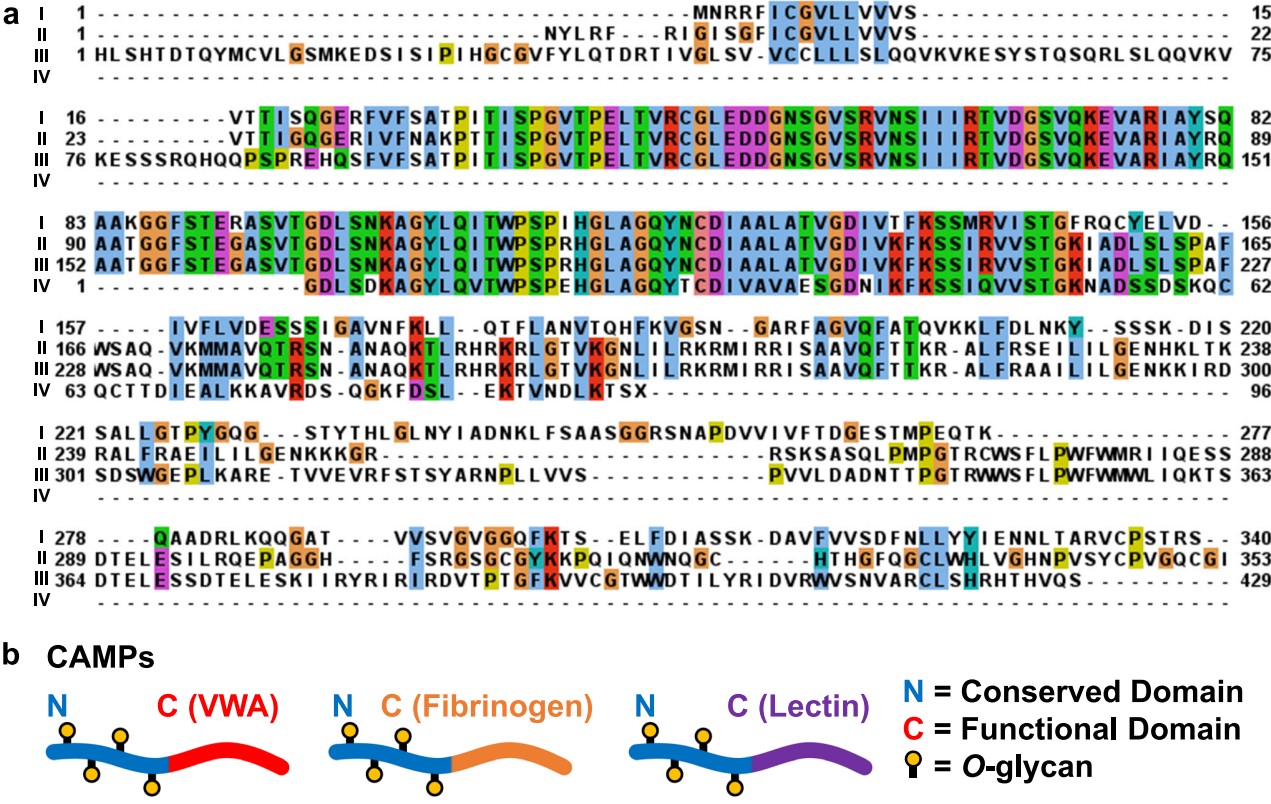

**Fig. 3 | Sequence analysis of CAMPs reveals similarity among *N*-terminal domains with interchangeable *C*-terminal functional domains. a** Multiple sequence alignment between (I) *C. aspserum* mucus VWA protein, (II) CAMP1 (truncated at *C*-terminus), (III) CAMP2, IV) CAMP3. **b** Schematic of CAMP architectures, showing conserved *N*-termini but varied *C*-termini between proteins.

collagen domains[59]. Novel4A was found to be Ser- (19.9%), Thr- (11.1%), and Pro-rich (12.7%), and domain analysis of this protein revealed similarity to proline-rich extensin glycoprotein signatures, which are involved in crosslinking aromatic amino acids in plant cell walls. To further investigate the structures of these proteins, glycoproteomic site-mapping studies using tandem mass spectrometry on trypsin-digested peptides were employed to detect glycopeptides from the isolated mucus protein samples and match these sequences to proteins isolated from the 260 kDa band (Supplementary Table 6). Novel4 released one glycopeptide modified with a HexNAcHex. Collagen5 presented six glycopeptides across all three mucus, three of which having two glycoforms, with the other three having one. Collagen5 also displays (Me)Hex$_3$HexNAc, (Me$_2$)Hex$_2$HexNAc, (Me)Hex$_2$HexNAc, Hex$_2$HexNAc, and HexHexNAc. Collagen7 was found to have one glycopeptide modified with (Me)Hex$_2$HexNAc. *N*-glycans were not detected on these proteins.

### CAMPs, a new class of mollusk proteins

Three proteins, CAMP1, CAMP2, and CAMP3, were identified that could not be readily categorized into the aforementioned groups. Proteomic analysis on SDS-PAGE-separated mucus proteins identified CAMP2 and CAMP3 only in the low molecular mass (<40 kDa) bands, consistent with the putative transcript sizes determined by RNAseq (Supplementary Fig. 15). Interestingly, CAMP1 was not detected in the SDS-PAGE, the reason for which requires further analysis. Upon multiple sequence alignment and BLAST and HMMER searches, these CAMPs were found to share sequence identity with each other and previously found, but not well-characterized, mollusk proteins[60]. These mollusk proteins from the databases contain lectin, VWA, and fibrinogen domains. The database proteins also share a general architecture with the three CAMPs identified here (Fig. 3a, Supplementary Table 7). We deem this class of proteins as 'CAMPs,' or Conserved Anterior Mollusk

Proteins because their *N*-terminal regions were nearly identical, but had entirely different *C*-terminal regions (Fig. 3b). These proteins' *N*-termini were abundant in Ser/Thr for potential glycosylation, had Ca$^{2+}$-binding pockets, and had oligomerization domains[61] that are irregularly spaced throughout the protein sequences. The *C*-terminal regions of CAMP1 and CAMP3 were fibrinogen-like domains and CAMP2 contained a Gal-specific lectin domain. From sequence alignment analysis, CAMP1 and CAMP3 were clustered with *C. aspersum* mucus VWA protein and a fibrinogen. CAMP2 was clustered separately, as it was paired alongside a possible H-type (GalNAc-specific) lectin. Glycoproteomic site-mapping studies using tandem mass spectrometry on trypsin-digested CAMP peptides revealed that CAMPs are modified with several *O*-glycans (Supplementary Table 8). CAMP1 presented one glycopeptide containing a FucHex$_2$HexNAc modification. CAMP2 presented four glycopeptides, one with two glycoforms modified with (Me)Hex$_2$HexNAc and HexNAc, while the other three peptides contained FucHex$_2$HexNAc, HexNAc$_2$, and HexNAc. One glycopeptide was detected on CAMP3 with two glycoforms, modified with (Me$_2$)Hex$_2$HexNAc and Hex$_2$HexNAc, respectively. No *N*-glycans were detected on CAMPs that met the criteria for identification. It is possible the *N*-terminal domains are involved in physically integrating the proteins into the mucus gel through noncovalent linkages, such as H-bonds, disulfide, and ion bridges with mucin proteins, while the *C*-terminal domains provide protein functionality.

### Glycomic analysis of *C. aspersum* mucus

*O*-glycans in mucins, which are *O*-linked to Ser or Thr via GalNAc residues, have been associated with lubrication, biological recognition, and network formation[62]. As such, the *O*-glycan compositions of the three mucus were individually analyzed. *O*-glycans from the snail mucus proteins were extracted by *β*-elimination with sodium borohydride[63], and their structures and abundances were identified

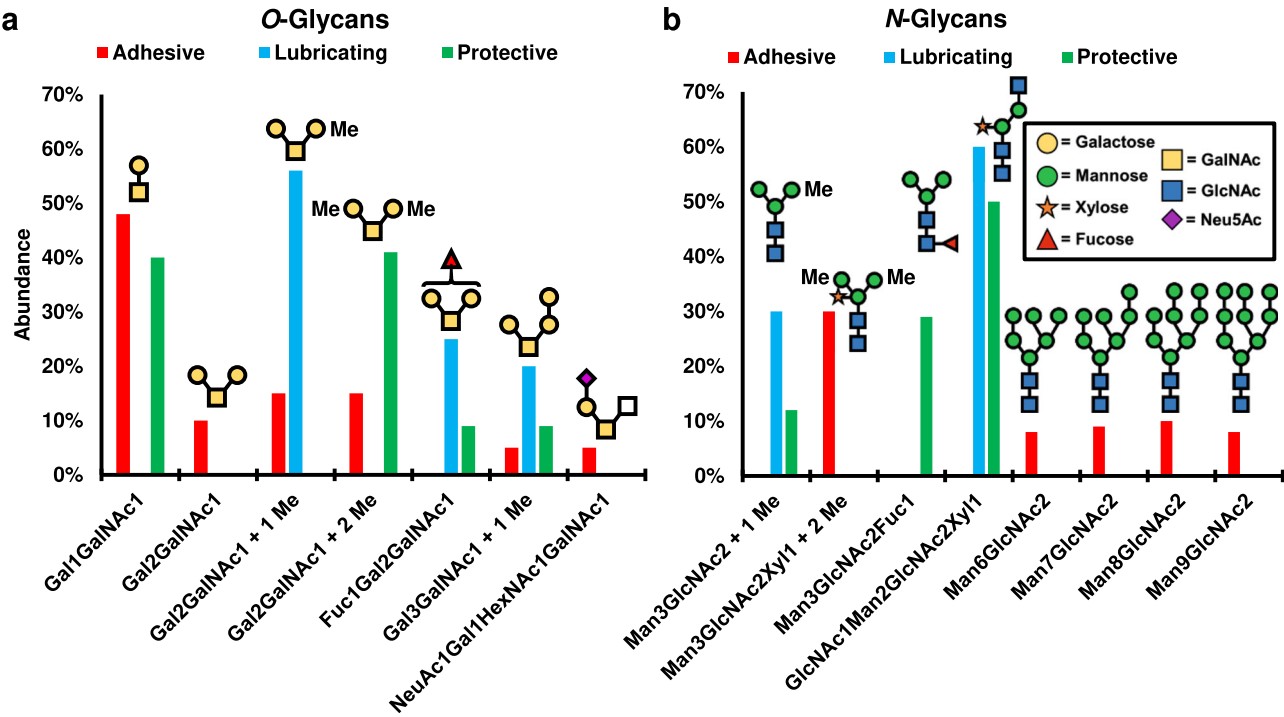

**Fig. 4 | Glycan content of snail mucus.** Structures and relative abundance of **a** *O*-glycans and **b** *N*-glycans found in each isolated mucus secretion by glycomic mass spectrometry. Glycans shown comprised >5% of glycomic abundance. Trace *O*- and *N*-glycans (<5% abundance) are listed in Supplementary Tables 9 and 10, respectively. Inset in 'b' lists monosaccharide structures as defined by the symbolic nomenclature for glycans (SNFG). Red: adhesive; blue: lubricating; green: protective.

using permethylation and MALDI (matrix-assisted laser desorption/ionization) mass spectrometry (Supplementary Figs. 16–18, Supplementary Table 9, Fig. 4a)[64]. Experiments were conducted using both iodomethane and iodomethane-D3 to identify native methylation[65].

Identified glycans are consistent with previous reports on mollusk glycosylation[64]. In the adhesive and lubricating samples, the mucin core-1 *O*-glycan (T-antigen) was the dominant glycan (52% and 69%, respectively). In the protective sample, the most abundant glycan was the trisaccharide (MeGal)$_2$GalNAc (48%), which was the second most-abundant glycan in all other samples. The T-antigen *O*-glycan ((Gal)GalNAc) was the second most abundant glycan in the protective sample. The trisaccharide (Gal)$_2$GalNAc and its methylated variants were observed in all three secretions. Adhesive and lubricating mucus contained the unmodified, mono-, and dimethylated versions of this glycan, but the protective mucus only contained the dimethylated form. The preponderance of so few glycans in the samples is surprising, given that human mucin glycans are diverse and typically utilize longer oligosaccharides[66]. Interestingly, the lubricating mucus showed sizable abundance (~8%) of several larger glycans, up to *n* = 5, while the other mucus possessed only trace amounts of these larger sugars. Gal-rich glycans have a recognized role in biological lubricity[67], and increases in polysaccharide length are accompanied with increased material stiffness[68]. Therefore, longer, galactose-presenting oligosaccharides have been observed in biological lubricants and may increase the stiffness of these secretions. Two sialylated *O*-glycans, Neu5AcGalGlcNAcGalNAc and Neu5AcFuc$_2$GalGlcNAcGalNAc, both of which were only found in the adhesive mucus, account for ~7% of this sample.

*N*-glycans were extracted from the proteins by treatment with PNGase F and identified by MALDI mass spectrometry (Fig. 4b, Supplementary Figs. 19–21, Supplementary Table 10)[69]. Twenty-four unique *N*-glycans were detected across all mucus samples. Compositions of these glycans are mainly consistent with *N*-glycans reported in mollusks[70]. The primary *N*-glycans identified in lubricating mucus are (MeMan)Man$_2$GlcNAc$_2$ and (Xyl)GlcNAcMan$_2$GlcNAc$_2$ oligosaccharides[70]. Protective mucus contained these two structures in addition to significant proportions of Man$_3$GlcNAc$_2$Fuc and oligomannose sugars. Adhesive mucus contained fifteen unique *N*-linked oligosaccharides not found in the other mucus secretions, with nine containing sialic acids. Six sialylated sugars were found in adhesive mucus, comprising 17% of the glycomic abundance, while three were observed in the protective mucus, comprising 6% of the abundance. No sialic acids were detected in the lubricating mucus. Neu5Gc was only detected in adhesive mucus. Interestingly, di- and tri-sialylation was observed exclusively in glycans of the adhesive mucus.

Overall, galactose, GalNAc, mannose, GlcNAc, fucose, xylose, Neu5Ac, and Neu5Gc were identified in the *O*- and *N*-glycans of snail mucus. Fucosylated structures were identified in the *N*-glycans of all three samples; however, only the protective and lubricating mucus contained fucosylated *O*-glycans. This observation supports previous accounts of low levels of fucosylation in invertebrate *O*-glycans[71]. The presence of methylation in both the *N*- and *O*-glycans supports prior reports on both marine and land snails, and identified compositions are similar to those identified in other snails[72]. A difference is the degree of methylation, as previous studies on other species reported high levels (3–4 methyl groups per monosaccharide), while our studies indicate lower levels of methylation (0–2 methyl groups)[64].

The purified mucuses were subjected to tryptic digestion and tandem mass spectrometry to identify mucus glycopeptides as well as verify glycan assignments (Supplementary Tables 11–13)[73]. Characteristic HexNAc, Neu5Ac, and Neu5Gc m/z peaks confirmed the presence of these monosaccharides in all three mucus samples (Supplementary Figs. 22–24). To further validate the presence of sialic acids on mucus proteins, fragmentation patterns of Neu5Ac-containing glycopeptides were detected through the MS/MS, confirming that snail mucus proteins are modified with sialic acids (Supplementary Figs. 25–27). While low-abundance oxonium ions for Neu5Gc were detected in the tandem

**Table 1 | Elemental composition of the snail mucus identified by energy dispersive X-ray spectroscopy (EDX) analysis**

| Element | Adhesive (wt%) | Lubricating (wt%) | Protective (wt%) |
|---------|----------------|-------------------|------------------|
| C | 58.0 | 79.5 | 62.2 |
| O | 15.6 | 8.29 | 18.8 |
| N | 9.38 | 4.61 | – |
| Cl | 4.50 | 1.15 | 5.87 |
| K | 4.11 | 3.00 | 2.55 |
| Na | 2.96 | 0.46 | 3.57 |
| Ca | 1.93 | 0.92 | 3.32 |
| S | 1.80 | – | 1.53 |
| Mg | 0.90 | 1.15 | 1.53 |
| P | 0.64 | – | 0.26 |
| Si | – | 0.92 | 0.38 |

Wt% refers to the percent abundance of each elemental species divided by its atomic molecular mass and normalized.

mass spectra, MS/MS analysis could not detect Neu5Gc-modified glycopeptides.

### Electron microscopy and elemental analysis of *C. aspersum* mucus

Mucus microscale morphologies and elemental composition were determined with SEM and EDX analysis, respectively, on fresh mucus deposited directly onto imaging substrates. The adhesive mucus formed large amorphous masses and ferning patterns that are consistent with mucus secretions in other organisms (Supplementary Fig. 28)[74]. Snail lubricious mucus appears oriented into thinner parallel lines. Protective mucus formed sheets that extended across much larger lengths than the other secretions. The elemental analysis of each mucus sample revealed substantial differences (Table 1, Supplementary Figs. 29–34). The lubricating mucus contained a higher carbon, oxygen, and nitrogen (organic) content (92.4%) compared to the adhesive (84.7%) or protective (82.52%) secretions. Of particular interest is the amount of $Ca^{2+}$ present in each mucus, as increased $CaCO_3$ content has been linked to increased mucus crosslinking and adhesiveness[4]. $Ca^{2+}$ appears to be present in varying amounts, and was measured to be 0.92%, 1.93%, and 3.32% in the lubricating, adhesive, and protective samples, respectively.

### Measurement of adhesive energy and elastic modulus

Stiffness and adhesion of the mucus hydrogels were characterized with scanning probe analysis. AFM imaging revealed that the adhesive and protective mucus were composed of large aggregates or sheets, while the lubricating mucus contained regions of smaller particles evenly spread across the surface (Supplementary Fig. 35). Similar sizes and morphologies were observed for the samples under ambient atmospheric conditions in the AFM and under vacuum in the SEM, suggesting that mucus morphology appears to be resilient to extreme changes in pressure and to desiccation. As such, the cracks observed in the imaging are not likely to be artifacts of the drying process.

AFM nanoindentation spectroscopy determined the mechanical stiffness and energy involved in mucus adherence. As mucus hydrogels are sensitive to moisture conditions, 50% relative humidity was maintained during the experiments by monitoring the chamber with a humidity sensor and injecting dry or moist air as needed to prevent fluctuations in gel swelling[75]. Since mucus is a heterogenous material, each sample was subjected to multiple indentations across different regions of the substrate ($n = 36$ for adhesive, $n = 51$ for lubricious, and $n = 59$ for protective) to account for topographical differences, and each indentation produced an approach-retraction force–separation

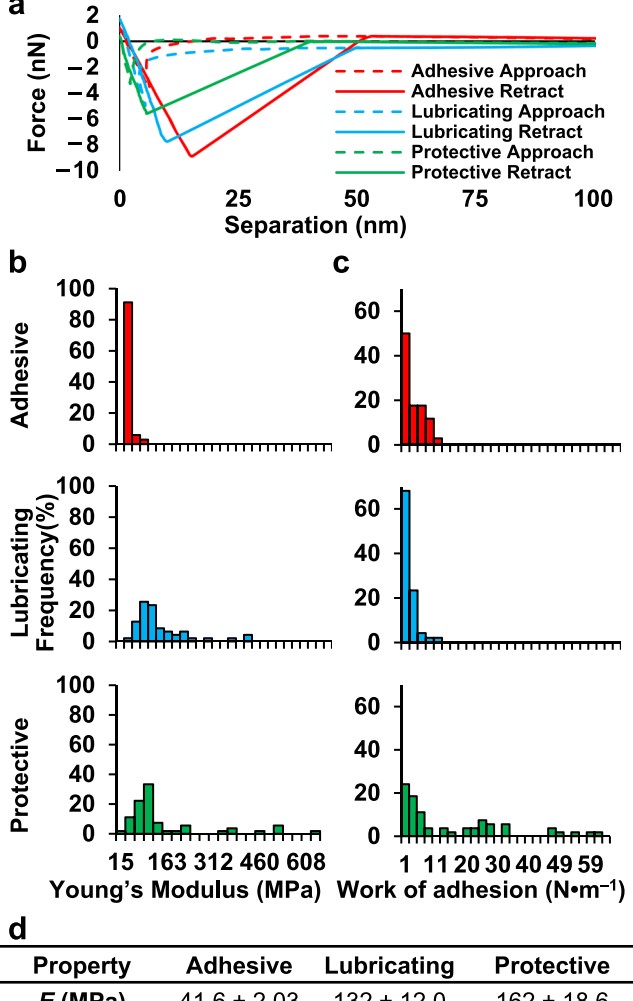

**Fig. 5 | Mechanical properties of adhesive (red), lubricating (blue), and protective (green) snail mucus determined by force-ramp indentation.**
**a** Representative force–distance curve pairs from the adhesive, lubricating, and protective snail mucus. **b** Distribution of measured Young's Modulus, *E*, and (**c**) work of adhesion, *W*, values for each mucus secretion. **d** Average values of *E* and *W* for adhesive ($n = 36$), lubricating ($n = 51$), and protective ($n = 59$) mucus from the data plotted in (**b**) and (**c**). Error is defined as standard error, or standard deviation divided by the square root of the number of measurements, *n*. Red: adhesive; blue: lubricating; green: protective. Source data are provided as a Source Data file.

| Property | Adhesive | Lubricating | Protective |
|----------|----------|-------------|------------|
| *E* (MPa) | 41.6 ± 2.03 | 132 ± 12.0 | 162 ± 18.6 |
| *W* (N·m⁻¹) | 3.37 ± 0.498 | 2.39 ± 0.268 | 17.2 ± 2.63 |

curve (Fig. 5a). Curves were fit using the Johnson–Kendall–Roberts (JKR) model[76], which accounts for adhesive interactions between the AFM tip and the sample. Curve fittings were then used to calculate Young's modulus (*E*) and the adhering energy (*W*) for each indentation[75]. The distributions of *E* (Fig. 5b) and *W* (Fig. 5c) across all indentations were determined, and average values for each distribution were calculated (Fig. 5d). Secreted mucus from other invertebrates have reported *E* of 0.1–200 MPa, and all the measured values fall within this range[77,78]. *E* for the adhesive mucus was significantly lower (41.6 ± 2.03 MPa) than that of the lubricating (132 ± 12.0 MPa) and protective (162 ± 18.6 MPa) samples, indicating adhesive mucus is much less stiff. Mucus from gastropods and other animals have *W* of 2–20 N·m⁻¹[79]. *W* for protective mucus (17.2 ± 2.63 N·m⁻¹) was much greater than adhesive (3.37 ± 0.498 N·m⁻¹) and lubricating (2.39 ± 0.268 N·m⁻¹) samples. The greater adhesiveness of protective mucus could increase this secretion's ability to trap pathogens and other materials.

## Discussion

Comparative analysis of these datasets revealed the origins of the adhesive, lubricating, and protective behavior of *C. aspersum* mucus (Fig. 1b). For example, adhesive mucus has low $E$ and high $W$ relative to the other secretions, which are desirable properties in biological adhesives[80]. Adhesive mucus can stretch and self-heal because of the H-bonds, ion bridges, and disulfide linkages that form reversible crosslinks, allowing dynamic sol-gel transitions without sacrificing material properties[81]. The gel's ion content also explains its flexibility. Hydrogels containing ionic crosslinks relieve stress through ion bridge reformation (stress relief occurs on a timescale of $t \sim 20$ s), while covalently crosslinked hydrogels employ water migration to relieve stress ($t \sim 1$ h), suggesting ion-dependent gels can stretch and spread more readily under mechanical stimuli[82]. Adhesive mucus contains large amorphous masses, which could increase contact surface area, thereby increasing adhesion between the snail and substrate. Greater glycoprotein[83], oligomeric protein[84], and glue protein[47] expression relative to the other mucus also contribute to the material's adhesiveness. Also, the adhesive mucus exclusively contains tyrosinase, which catalyzes the formation of DOPA, a chemical signature of adhesives secreted by mussels, which is possibly why enzymes involved in its metabolism are found in adhesive mucus[47]. DOPA-based adhesion has not been reported in snails, and this finding suggests that *C. aspersum* could utilize a similar mechanism as mussels and other mollusks to adhere to inorganic substrates[47]. Elemental analysis determined this mucus has comparatively high $Ca^{2+}$ content. Secretion of $Ca^{2+}$ likely increases mucus adhesion by coordinating ion bridges in the hydrogel[4,22]. This idea is further supported by the presence of acidic sialylated glycans, which would increase cation binding within the gel and to substrates. As only the adhesive mucus contains Neu5Gc, it is possible only pedal tissue expresses Neu5Gc-synthesizing enzymes, while the dorsal tissue does not, supported by the fact that Neu5Ac is a biosynthetic precursor of Neu5Gc[85]. Together, these chemical compositions may generate weak cross-linking that constructs a continually reforming flexible adhesive, allowing the snail to adhere to the roughest horizontal, inclined, and inverted surfaces.

*C. aspersum* lubricious mucus was stiff and minimally adhesive, and these properties may provide minimal friction and adhesion on the snail's pedal surface. Linear structures along its axis of motion reduce surface roughness and, in turn, friction[86]. A large portion of this hydrogel's composition (~40%) was collagen. Increased collagen levels increase hydrogel stiffness[87], which explains the elevated $E$ of the lubricious mucus compared to the adhesive (<10% collagen). Compared to the Ca-rich adhesive mucus, hydrogels formed with cross-linked collagen, like the lubricious mucus, require longer timescales to relieve mechanical stress and thus would have increased $E$[82]. $Ca^{2+}$-binding proteins were abundant in lubricating mucus, which may sequester free $Ca^{2+}$ and prevent ion bridges from forming. Elemental analysis revealed the gel's $Ca^{2+}$ content (0.92%) was less than half of that in the adhesive (1.93%) and one-third the amount in the protective (3.32%). Lower salt concentration leads to fewer ionic crosslinks in the gel, resulting in lower $W$ of the lubricious mucus[4]. The elongated oligosaccharides found in the lubricious mucus are extensively hydrated and minimize glycan chain interpenetration under low loads, like those experienced by the snail, which is known to increase lubricity[62]. The result of all of these elements is a rigid non-stick gel underneath the snail during locomotion, allowing effortless movement across any surface.

*C. aspersum* protective mucus combines features from the adhesive and lubricative mucus, resulting in a hybrid material with the stickiness of the adhesive mucus and the rigidity of lubricious mucus. Protective mucus forms contiguous sheets covering more surface area than the other secretions. Like the adhesive, the protective mucus shows high $Ca^{2+}$ content alongside glue proteins and glycoproteins, thereby potentially increasing $W$. These proteins are modified with short *O*-glycans, which could decrease lubricity. Like the lubricious mucus, the protective mucus has high collagen expression (~25% of protein expression), which correlates to high $E$. The stiffness may also be the consequence of ionic, covalent, and noncovalent linkages forming within the hydrogel[81,82], which could allow the gel to relieve stress via both ion bridge rearrangement and water migration[82], as the role of $Ca^{2+}$ in forming snail mucus ion bridges is well known[9]. Consequently, mechanical inputs would have their energy dispersed across different timescales. The protective gel would thereby respond to stress more quickly than covalently linked hydrogels but also experience more strain than ionically crosslinked ones. Additionally, these behaviors could indicate a tougher gel that can better maintain its integrity. Though all three mucus contained protease inhibitors, which shield host proteins from degradation[51], the protective hydrogel's distinguishing factor is the presence of lectins, which confer antimicrobial properties to the mucus by preventing pathogenic binding[54]. This material also possesses diverse *N*-glycans, which likely increase interactions with pathogens in the environment, thereby trapping threats and allowing defensive proteins to engage.

The comparative analysis of the three distinct mucus illuminates the origins of their functional differences (Fig. 1b). General principles regarding snail mucus were elucidated, leading to important findings that can be used to advance the field of mucus research. Secreted snail mucus are majorly comprised of collagens and glycoproteins. A relatively simple set of 8 *O*-glycan structures decorate these *C. aspersum* mucus proteins compared to, for example, 76 in *Xenopus laevis*[88] and 169 in *Salmo salar*[89]. The *O*-glycan length is modulated between mucus, possibly altering stiffness and lubricity[62]. Each mucus has drastically different *N*-glycosylation. Sialic acids, which are uncommon in mollusks[72], were detected in *O*- and *N*-glycans of the adhesive mucus. Covalent, noncovalent, and ionic crosslinking appears to have a significant effect upon mechanical properties, and snails rely upon defensive proteins to protect hydrogel integrity. Additionally, CAMPs containing *N*-terminal glycodomains and *C*-terminal functional domains were identified, showing there is much to be learned by identifying and annotating mucus genes. The comparative mucomics strategy applied here for *C. aspersum* can be used to determine how compositions of other animal secretions account for their ecological function or to assist in the development of synthetic analogues with similarly advantageous biological and chemical properties[90,91]. It must be acknowledged that there are limitations to this mucomics approach, including that it cannot prove the exact contribution any individual structural component has on hydrogel properties. However, this holistic approach, where protein, glycan, and ion content are all considered, is the most promising path towards understanding how the various components contribute to the properties of these complex, heterogeneous biomaterials.

## Methods

### Materials

All chemicals were purchased from VWR unless otherwise noted.

### Mucus collection

Snails were provided in October 2021 by Peconic Escargot (Cutchogue, NY, USA), where they were cultured at room temperature and provided a diet of dirt, wild herbs, and cultivated herbs *ad libitum*. 25 physically active snails that were between 5 and 7 cm were washed with room temperature tap water to remove food, debris, and pathogens and placed into a plastic aquarium. Snails were allowed to crawl freely on petri dishes to collect lubricating mucus. To collect adhesive mucus, snails were placed against an inverted dish until adhered and left suspended for 15 min. Lubricating and adhesive mucus were not processed or manipulated further and were immediately placed on ice for preservation. Protective mucus secretion was induced by gently rubbing the snail's back with a spatula, which was scraped off into a

plastic test tube. All mucus samples were stored under ice packs without further processing in an insulated cooler for transport to the laboratory, where they were then stored at –80 °C until use.

## Mucus protein purification

Mucus samples were thawed and physical debris was removed with tweezers. 2 mL of 6 M Guanidinium HCl (Gdn), CsCl (density 1.388 g/mL) was added to mucus-containing petri dishes and incubated at 4 °C overnight to dissolve mucus. Additional residue was collected from the dishes by gently scraping residue with a razor blade. Mucus-containing solutions in the petri dishes were pooled by mucus type into 13.2 mL ultracentrifuge tubes (Beckman-Coulter). Samples were then subjected to isopycnic density gradient ultracentrifugation[92] in a swinging bucket SW41 Ti rotor ultracentrifuge (35,000 rpm, 72 hr, 4 °C) at a relative centrifugal force of 150,000 × g, within which mucus migrates to a characteristic band and cells are removed from the solution. Following centrifugation, tubes were pierced with a needle and fractionated (0.5–1 mL). Each fraction was measured for density and tested for carbohydrate content using a microtiter periodic acid-Schiff's reagent (PAS) staining protocol[93]. Fractions with a density of approximately 1.4 g/mL as well as high signal-to-background absorbance at 550 nm, indicating high glycoprotein content, were considered mucus-positive. Mucus-positive fractions were pooled and dithiothreitol (DTT) was added to each pool to reach a final concentration of 0.05 M DTT and shaken at 45 °C overnight in an Echotherm orbital mixing dry bath (Torrey Pines Scientific) to reduce disulfide bonds in the mucus hydrogel networks. Reduced samples were then dialyzed (MM cutoff 2 kDa) against 3 changes of ultrapure water over 48 h and fluffy white precipitate formed. Samples were then lyophilized at –55 °C/1 mbar, resulting in a light beige powder which was stored at –80 °C. Protein content was quantified at each step in the purification using a Nanodrop one-C spectrophotometer (Thermo-Fisher).

## SDS-PAGE

Mucus samples were collected from snails in January 2023 in the same manner as described in the "Mucus Collection" section. Purified mucus proteins were solubilized in 1% (v/v) Tween solution and mixed with 2X SDS loading buffer (Quality Biological, 351-082-661). Samples were vortexed and incubated at 95 °C for 15 min before loading in triplicate on a 4–20% gradient gel. Electrophoresis was conducted at 150 V for 54 min. Following electrophoresis, gels were divided into three identical sections and each was subjected to Coomassie, Silver, or PAS staining, respectively (Supplementary Figs. 9–11). Coomassie gel lanes were divided into three slices to obtain molecular mass ranges of <40 kDA, 40–150 kDA, and 150+ kDa. Slices were stored on ice packs prior to downstream proteomic analysis.

## RNA extraction and sequencing

Snails provided by Peconic Escargot in February 2020 were sacrificed on-site via freezing in a dry ice-ethanol mixture. Whole snails were stored in Invitrogen RNAlater™ (Thermo Fisher, AM7021) and frozen at –80 °C until used. 6 individual tissue slices of the snail's dorsal and pedal surfaces of the foot were excised from different snails. Total RNA was extracted from these slices using a Qiagen RNeasy Micro kit (Qiagen, 74004) according to manufacturer's instructions. The integrity of total RNA was confirmed using nanodrop and Agilent 2100 BioAnalyzer analysis. The RNA Integrity Number (RIN) was not considered because of known co-migration of 28S rRNA fragments with 18S rRNA in molluscan RNA, causing decreased RIN values in the absence of RNA degradation[94,95]. Total RNA was used as a template to perform polyA enriched first strand cDNA synthesis using the HiSeq RNA sample preparation kit for Illumina Sequencing (Illumina Inc., CA) following the manufacturer's instructions. The cDNA libraries were sequenced using Illumina HiSeq 1000 technology using a paired end flow cell and 80 × 2 cycle sequencing.

## Read processing and De Novo assembly

Raw reads were quality checked with FastQC v0.11.5 (www.bioinformatics.babraham.ac.uk)[96]. Adapter sequences and low-quality reads (Phred score <33) were removed using Trimmomatic v0.36 and trimmed reads were re-evaluated with FastQC to ensure the high quality of the data after the trimming process[97]. Due to the lack of a reference genome, the processed reads were de novo assembled using Trinity v2.4.0[98]. De novo assembled transcriptomes were translated with Trinity Super Transcripts[31]. Supertranscripts was used to construct the largest isoform of each gene, in other words producing the original unspliced transcripts, rather than spliced variants of the transcripts[31]. 179,552 transcripts were assembled. RNA sequences were deposited in Genbank with the primary BioSample accession codes SAMN29856567, SAMN29856568, SAMN29856569, SAMN29856570, SAMN29856571, SAMN29856572, and SRA accession codes SRR20337023, SRR20337022, SRR20337021, SRR20337020, SRR20337019, SRR20337018.

## Proteomic mass spectrometry

2 µg of purified snail mucus protein samples at a concentration of 1 mg/mL were loaded onto a single 10% SDS-PAGE stacking mini gel (#4561034, BioRad) band to remove lipids, detergents and salts. The single gel band containing all proteins was reduced with DTT, alkylated with iodoacetic acid and digested with trypsin. 2 µg of extracted peptides were re-solubilized in 0.1% aqueous formic acid and loaded onto a Thermo Acclaim Pepmap (Thermo, 75uM ID X 2 cm C18 3uM beads) precolumn and then onto an Acclaim Pepmap Easyspray (Thermo, 75uM X 15 cm with 2 µM C18 beads) analytical column separation using a Dionex Ultimate 3000 uHPLC at 250 nL/min with a gradient of 2–35% organic (0.1% formic acid in acetonitrile) over 1 h. Peptides were analyzed using a Thermo Orbitrap Fusion mass spectrometer operating at 120,000 resolution (FWHM in MS1) with HCD sequencing (15,000 resolution) at top speed for all peptides with a charge of 2+ or greater.

## Bioinformatic analysis

The raw data were converted into *.mgf format (Mascot generic format) for searching using the Mascot 2.6.2 search engine (Matrix Science) against predicted sequences from the de novo assembled snail transcriptome[99]. The database search results were loaded onto Scaffold Q+ Scaffold_4.9.0 (Proteome Sciences) for statistical treatment and data visualization[100]. Peptide identifications were made by exact homology of fragmented peptides against translated transcripts. Using the Scaffold Local FDR (false discovery rate) algorithm, probability thresholds for peptide identifications and protein identifications were set at 95.0% and 5.0%, respectively, to achieve an FDR less than 1.0%, as per proteomic research standards[19,30]. Additionally, accepted sequences must have contained at least 2 identified peptides. Peptides were quantified by MS/MS counts. The mass spectrometry proteomics data have been deposited to the ProteomeXchange Consortium via the PRIDE partner repository with the dataset identifier PXD035534 and 10.6019/PXD035534[101].

The sequences of the proteins identified in the mucus samples were subjected to BLASTP[58] searches using default parameters to determine their functions based on homology with known proteins in the NCBI non-redundant protein database[16]. Simultaneously, HMMER[60] was also used to conduct domain searches against the PFAM[59] protein database. Employing these two platforms in tandem results in more accurate functional assignments based upon sequence identity and shared domain structures. Each protein was manually classified into one of nine functional categories: lectin, glycoprotein, network formation, matrix, enzymes, protease inhibitors, ion-binding, regulatory, or housekeeping. Proteins that had sequence similarity with predicted snail proteins without known function were classified as "Unknown," and proteins that had no similarity with any known

proteins were classified as "Novel." Using Clustal Omega[102] within the EMBL−EBI web form (ebi.ac.uk/Tools/msa/clustalo/) and using default parameters, a multiple sequence alignment was conducted on our 71 proteins as well as an extensive set of reference proteins to generate a dendrogram and cluster the genes studied via neighbor-joining. For each protein type found, 3–5 proteins of the same type from other gastropods or mollusk species were selected from the NCBI non-redundant protein database and added to the alignment. Additionally, other protein types that appeared in the initial BLAST search results were included to build more accurate relationships. By using a global comparative approach[103], validation of protein functional assignments and characterization of the more elusive proteins are streamlined. In most cases, proteins of a given type were paired alongside known proteins of the same type, with only minimal cases of orphaned sequences. Molluscan proteins of each functional category, as well as three human mucins, were included in the tree generation. Three proteins of an unrelated family were included as an outgroup. Display and annotation of alignment tree were conducted using iTOL v5[104]. Sequences were uploaded into the HMMER web server for identification of domains[60]. Multiple sequence alignment of proteins was conducted using Jalview[105].

### Glycoproteomic tandem mass spectrometry

Purified mucus protein samples ($n = 1$, 3 biological replicates) were resuspended in 100 μL of 50 mM ammonium bicarbonate. To this, 100 μL of 25 mM DTT was added. Samples were vortexed and incubated at 45 °C for 45 min. Following incubation, sample were allowed to cool to room temperature, and 100 μL of 90 mM iodoacetamide (IAA) was added. Samples were vortexed and incubated at room temperature in the dark for 20 min. Following incubation, samples were cleaned with 3 kDa MWCO filters (Millipore Amicon Ultra, UFC500396). Prior to loading samples, the filters were washed twice with 400 μL of 50 mM ammonium bicarbonate. Filters were spun for 10 min at 14,000 rpm. Following washing, samples were loaded onto the filters and spun at 14,000 rpm for 25 min. 400 μL of 50 mM ammonium bicarbonate was added, and samples were spun once more. This process was repeated one more time. The desalted protein sample was removed from the filter by inverting it into a clean tube and centrifuging for 5 min. The filters were rinsed with 50 μL of 50 mM ammonium bicarbonate, and the samples were inverted once more and centrifuged. The total volume was then brought to 100 μL. A 5 μL aliquot was digested with 1 μg of trypsin (Promega) overnight at 37 °C. Trypsin was then terminated by heating samples to 100 °C for 5 min. Samples were then passed through a 0.2 μm filter. Samples were diluted to 30 μL in 0.1% formic acid.

Samples were analyzed using a Thermo Fisher Eclipse Tribrid mass spectrometer equipped with a nano electrospray source and coupled to an Ultimate 3000 RSLCnano liquid chromatography system. Samples were analyzed using a 180-min gradient. A prepacked nano-LC column of 15 cm length and 75 μm internal diameter, filled with 3 μm C18 material was used. The precursor ion scan was acquired at 120,000 resolution in the Orbitrap, and precursors with a time frame of 3 s were selected for MS/MS fragmentation in the Orbitrap at 15,000 resolution. Monoisotopic precursor selection was selected and the threshold for MS/MS triggering was 1000 counts. MS/MS fragmentation was done using stepped higher energy collision-induced dissociation (HCD) product triggered collision induced dissociation (CID) (HCDpdCID). Precursors with an unknown charge state or charge state of +1 were excluded, and samples were run in positive ion mode[73].

LC-MS/MS spectra were searched against the FASTA sequences of the Helicidae genome obtained from Uniprot, as well as the protein sequences obtained from proteomics experiments and RNAseq experiments outlined in this paper. Byonic software (Ver 5.0.12) was used for data analysis. Oxidation of methionine, deamidation of asparagine and glutamine were searched as variable modifications and carbamidomethylation of cystine was searched as a fixed modification. A Byonic N-glycan database of insect and plant glycans as well as a tailored glycan list developed based on the glycomics results were searched as variable modification for N-glycans. A Byonic O-glycan database of 9 common O-glycans as well as a tailored glycan list developed based on our glycomics results were searched as variable modification for O-glycans. The data was then filtered based on a |log prob| value equal to or greater than 3, and a Delta Mod Score equal to or greater than 50[106]. Matches were manually verified by confirming presence of oxonium ions and expected neutral loss patterns.

To identify glycoprotein candidates attributed to prominent 260 kDa bands in all three samples, peptides were extracted and analyzed by LC-MS/MS on the same instrument for glycoproteomic characterization. Entire duty cycle was used for stepped HCD fragmentation for high confidence peptide backbone sequencing. Common contaminants and decoys generated by Byonic software were added to the FASTA databases to interrogate candidate components in the bands. Glycoproteomics data were submitted to GlycoPost database under the accession number GPST000297.

### Release of *N*-glycans

Lyophilized mucus protein samples were reduced by adding 25 mM DTT and incubating at 45 °C for 60 min. DTT was then removed using Amicon ultracentrifuge 10 kDa spin filters. Samples were then resuspended in 50 mM ammonium bicarbonate and 2 μL of PNGase F (New England Biolabs) was added. Samples were incubated at 37 °C for 48 h, and an additional aliquot of PNGase F was added after 24 h. Following incubation, released *N*-glycans were separated from the deglycosylated protein by passing through an Amicon ultracentrifuge 10 kDa spin filter. The flow-through was then loaded onto a C18 SPE cartridge (Resprep) and eluted with 5% acetic acid. The *N*-glycan fraction as well as the de-*N*-glycosylated protein fraction were lyophilized.

### Release of *O*-glycans

Dried samples were then dissolved in 0.1 M NaOH and mixed with 55 mg/mL sodium borohydride. Samples were then subjected to 52 hr β-elimination at 45 °C. Following incubation, samples were neutralized with 10% acetic acid dropwise and passed through DOWEX H+ resin column and C18 SPE column. Samples were eluted with 5% acetic acid. Eluted *O*-glycans were then lyophilized. Borates were removed using 9:1 methanol: acetic acid under a stream of $N_2$.

### Per-*O*-methylation and profiling by matrix-assisted laser-desorption time-of-flight mass spectrometry (MALDI-TOF-MS)

Dried samples ($n = 3$, 3 biological replicates) were then dissolved in dimethylsulfoxide (DMSO) and methylated using NaOH/DMSO base and methyl iodide. The reaction was quenched using LC-MS grade water, and Per-*O*-methylated glycans were extracted with methylene chloride and dried under $N_2$. Permethylated glycans were dissolved in methanol. Glycans were then mixed 1:1 with α-dihyroxybenzoic acid (DHB) matrix. MALDI-TOF-MS analysis was done in positive ion mode using an AB SCIEX TOF/TOF 5800 mass spectrometer on three samples. Glycans were identified according to previously established snail glycan assignments. Fetuin was processed identically as a glycoprotein control. 0.1 μg of xylotetraose (Megazyme) was added to each sample as a glycan control. Glycomics data were submitted to GlycoPost database under the accession number GPST000297.

### Scanning electron microscopy/energy dispersive x-ray spectroscopy

To create the samples, live snails were allowed to crawl on SEM Al pin stubs (Ted Pella, 16144) that were inverted or horizontal, to create samples for adhesive and lubricating mucus, respectively, and back (protective) mucus was scraped onto the stubs, similar to the silicon wafer samples for AFM, and air-dried overnight. The samples were

sputter-coated with gold to a thickness of 5 nm using a Leica EM ACE600 Coater for better electrical conductivity. These samples were then imaged in a Thermo Scientific (FEI) Helios NanoLab 660 FIB-SEM with HT of 5 kV, current of 6.3, 13, and 25 pA with ETD (Everhart-Thornley) detector. EDS (energy-dispersive X-ray spectroscopy) mapping was collected with an Oxford detector at HT of 10 kV and current of 1.6 nA. Data was collected and analyzed using AZtec software[107].

## AFM Topography

To create the samples, live snails were allowed to crawl on Si wafers that were inverted (adhesive) or horizontal (lubricating), while back (protective) was directly deposited onto the wafer and directly analyzed. The samples were subjected to AFM imaging and analysis using an AFM (Multimode 8, Bruker) under ambient temperature (25 ˚C) and relative humidity (50%) to mimic conditions experienced by snails in the wild. Mucus topographies were measured by using an AFM probe with a tip radius of ~2 nm (SCANASYST-AIR, Bruker). All measurements were taken at 50% relative humidity, which was controlled by injecting dry or moist air into the enclosed AFM chamber and measured by a humidity sensor (HIH-4021, Honeywell).

## Stiffness and work of adhesion characterization via the JKR model

The stiffness of mucus samples was characterized using AFM nano-indentation method[76], where an indenter (MLCT-E, Bruker) with a radius of 20 nm and a spring constant of 0.139 N/m was used (NanoScope Analysis 1.9). The indentation deflection sensitivity was 40.7 nm/V, calibrated by performing an indentation on the silicon wafer substrate. Peaks of three mucus aggregates are indented to obtain the force vs. displacement relationships, of which the retracting portion of the indenting profiles were subsequently analyzed by using the Johnson–Kendall–Roberts (JKR) model, given by

$$E_{JKR} = \frac{9\pi R^2 \Delta r}{2a_0^3},$$ (1)

$$P_{adh} = -\frac{3}{2}\pi \Delta r R,$$ (2)

$$h_t - h_0 = \frac{a_0^2}{R}\left(\frac{1+\sqrt{1-\frac{P}{P_{adh}}}}{2}\right)^{\frac{4}{3}} - \frac{2}{3}\frac{a_0^2}{R}\left(\frac{1+\sqrt{1-\frac{P}{P_{adh}}}}{2}\right)^{\frac{1}{3}},$$ (3)

where $E_{JKR}$ is Young's modulus, R is the tip radius, $\Delta r$ is the work of adhesion, $a_0$ is the contact area when the contact force is zero, $P_{adh}$ is the pull-off force, $h_t$ is the indentation depth, $h_0$ is the contact point where the pull-off force shows, and $P$ is the load. The work of adhesion was measured by the area enclosed by the approaching and the retracting indentation force–displacement curves, and was normalized by the probe sample contact area ($a_0$), given by

$$a_0 = \pi R h_t.$$ (4)

## Reporting summary

Further information on research design is available in the Nature Portfolio Reporting Summary linked to this article.

# Data availability

The mass spectrometry proteomics data generated in this study have been deposited in the ProteomeXchange Consortium via the PRIDE partner repository under the accession code PXD035534. RNA sequences that support the findings of this study have been deposited in Genbank with the primary BioSample accession codes SAMN29856567, SAMN29856568, SAMN29856569, SAMN29856570, SAMN29856571, SAMN29856572 and SRA accession codes SRR20337023, SRR20337022, SRR20337021, SRR20337020, SRR20337019, SRR20337018. FASTA files of all Trinity-assembled RNA sequences used, their translated amino acid sequences, and the annotated shortlist of the proteins cross-validated with proteomics can be found via FigShare with https://doi.org/10.6084/m9.figshare.28949312. Glycomics data and glycoproteomics data generated in this study were submitted to GlycoPost database under the accession number GPST000297. The NCBI non-redundant protein database and PFAM protein database were used in this work. Uncropped SDS-PAGE gel images are provided as a Source Data file. The AFM mechanical property data are provided as a Source Data file. The authors declare that all other data that support the findings of this study are available within the paper and its supplementary information files. Source data are provided with this paper.

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

## Acknowledgements

This work is supported by a CUNY Science Scholarship and a CUNY Llewellyn Fellowship (A.R.C.); the Air Force Office of Scientific Research grants FA9550-19-1-0220, FA9550-18-1-0142, FA9550-23-1-0230 and the National Science Foundation grant DMR-2212139 (A.B.B.); the Complex Carbohydrate Research Center (Athens, GA), the National Institutes

of Health-funded R24 grant NIH-R24GM137782 and NSF GlycoMIP, a National Science Foundation Materials Innovation Platform funded through Cooperative Agreement DMR-1933525 (L.E.P., X.Y., and P.A.); the Office of Naval Research N00014-18-1-2492 (X.C.); the Army Educational Outreach Program (Rochester, NY) and Harlem Educational Activities Fund (New York, NY) (D.B. and A.B.B). the Allen Institute's Distinguished Investigator Award and NIH SPEECH Pilot Project U54CA221704, U54CA221705 (M.H.). Taylor Knapp and Peconic Escargot (Cutchogue, NY) are acknowledged for providing the animals used in this study. Genevieve Arroyo, Nicholas Mueller, and Robert Gullery are acknowledged for assisting in the collection of snail mucus. The Gardner, Cassacia, and Ulijn labs at the CUNY Advanced Science Research Center are acknowledged for allowing use of lab equipment and space to conduct experiments. NYU's Genome Technology Center is acknowledged for conducting the transcriptomic sequencing with support from the Laura and Issac Perlmutter Cancer Center (Cancer Center Support Grant P30CA016087). The Clinical Proteomics Platform at the RIMUHC, McGill is acknowledged for conducting the proteomics experiments and analysis.

## Author contributions

A.B.B. conceived the research. A.B.B., A.R.C, M.H., X.C., L.E.P. and P.A. coordinated the research. A.R.C. and M.B.M. collected the snail mucus samples and snails used in these studies. A.R.C. conducted mucus purifications. L.E.P. extracted glycans and conducted glycomic analysis and glycoproteomic analysis. A.R.C. and M.B.M. conducted RNA extraction and transcriptomic analysis. Z.-L.L. conducted AFM experiments. A.R.C., Z.-L.L., and D.B. conducted AFM mechanical property analysis. A.R.C. and S.Z. conducted the SEM and EDX experiments. X.Y. conducted proteomic analysis. All authors contributed to analysis and discussion of data. The manuscript was written by A.R.C. and A.B.B. and edited and approved by all authors.

## Competing interests

The authors declare no competing interests.

## Additional information

**Peer review information** : *Nature Communications* thanks Niclas Karlsson, and the other, anonymous, reviewers for their contribution to the peer review of this work. A peer review file is available.

