## [Peer Review File · Nature Communications]

REVIEWER COMMENTS

Reviewer #2 (Remarks to the Author):

The report “Comparative Mucomic Analysis of Three Functionally Distinct Cornu aspersum Secretions” by Cerullo et al. describes the mucus characterization of various secretions from Snail. This includes transcriptomics, proteomics, glycomics and analysis of inorganic cations. The report appears to be an initial mapping of these mucus in order to guide the researchers into further discoveries. As such, the report does not provide any novel discoveries, or alternatively, the findings are on the level of interesting observation but is lacking the rigor required for being considered to be concluded. The structure function-relationship is on the level of literature reviewing to find relations, but the authors make very little efforts in trying to verify their findings. Also, the authors have not made any substantial alteration on their manuscript based on my suggestion, apart from some obvious mistakes in their figures and tables

Major issues.

The authors are claiming in the abstract that they have identified new proteins in the hydrogel what they called CAMPs, without any further explanation. It is not obvious from Figure 3 that all these CAMPs contain signalpeptide that allows them to be secreted. If I understand it correctly from figure 2, there was no proteomic identification of these molecules, either. In addition, since the authors claims that the mucins from the snail are quite small, it would have been nice to see SDS-PAGE stained with for instance PAS of their isolated fraction, in order to appreciate the difference between snail and mammalian mucins, of course it would be possible to perform proteomic also on individual bands. It would also be suggested that the authors are providing additional evidence of the size of the CAMPs and the sequence using for example pcr to determine the transcript of the proposed mucins.

The term “mucomic” is not very well defined, but to my understanding it would include identification of mucins in the snails mucus. From the dendogram in figure 2 only 9 mucins were identified by proteomics among the 50+ putative mucins in the samples. Strangely enough the authors claims that they only found 2 clusters of mucins, but the cluster in black on the left hand upper corner also contains “mucins”. This reviewer fail to find the CAMPs in the dendogram. Few of the “mucins” detected by proteomics are indeed recognized as mucins when they are found in other species. Very few of the proteins named as MUC was identified by proteomics Mucin type molecules are notoriously difficult to detect in

proteomics, indicating that there may be technical difficulties to identify what are the “mucins” that are actually present in the snail secretion. In general, there are very few of the proteins identified by transcriptomics that are detected in the mucus by proteomics. This indicates to me that there are technical issues concerning the “mucomics”. The authors may also justify why they need to introduce another -omics, and what this new term actually means. Maybe something like system mucus biology would make more sense?

Glycomics data:

For N-linked the authors reports that some structures contain only NeuAc while others only NeuGc. This is a strange result that is left without any explanation. Also, there are no attempt to verify the glycans for instance by exoglycosidases digestion or MS/MS.

Elemental analysis

All analysis (Supplementary figure S19-21) contains high amount of aluminum. One can expect that this is a contamination. With this high amount of contamination, the measurement of other metal ion is put in doubt.

Overall

In general, the paper is not providing a connection between macroscopic property and molecular characterization, and major “discoveries” in the report are not validated. (eg the identification of CAMPs, presence of DOPA or the connection between glycosylation and adhesiveness/protection/lubrication). The authors demonstrate that they have limited understanding of previous literature, where they claim that there are “longer oligosaccharides likely contribute to its lubricative properties, while the shorter O-glycans found in the adhesive and protective mucus would attenuate lubrication” even though the reference used show that short glycans are enough for providing lubrication

Minor issues:

Page 5. What is proteomic abundance?

Page 9 The sentences about L-dopa role in snail has not been substantiated in this report. Much of this is speculation and should be moved to discussion

REFeree 2

Comment 1: The report “Comparative Mucomic Analysis of Three Functionally Distinct Cornu aspersum Secretions” by Cerullo et al. describes the mucus characterization of various secretions from Snail. This includes transcriptomics, proteomics, glycomics and analysis of inorganic cations. The report appears to be an initial mapping of these mucus in order to guide the researchers into further discoveries. As such, the report does not provide any novel discoveries, or alternatively, the findings are on the level of interesting observation but is lacking the rigor required for being considered to be concluded. The structure function-relationship is on the level of literature reviewing to find relations, but the authors make very little efforts in trying to verify their findings. Also, the authors have not made any substantial alteration on their manuscript based on my suggestion, apart from some obvious mistakes in their figures and tables

Previous Response: We respectfully disagree with the Referee that this work does not report any novel discoveries, and acknowledge the words of Referee 1, who stated that “*the paper provides a well written and comprehensive investigation of a land snail mucus molecular components, to a level not achieved by other investigators to date. The outcomes are of interest to the areas of biomaterial science, snail ecology and pharmacology*”. And Referee 3, who stated “*This work is of high quality, it is reported in an appropriate way, and will be of obvious important ground work for the mucus field, and also surely for those interested in C. aspersum physiology. As such, it should be published after addressing some of the concerns listed below.*” Both Referees 1 & 3 recognize that the novelty and benefit of the manuscript comes from the unique comparative analysis, which gives us the unprecedented ability to understand how the different components work together to give rise to the different ecological functions of the adhesive, lubricating, and protective mucuses. This work identifies 32 glycans and 71 proteins found in snail mucus, 18 of which display no similarity to any previously reported proteins. We also identify a new family of proteins with homologs in other mollusks that have not yet been explored. Furthermore, we find that the presence of ions and collagen crosslinking are correlated with the material properties of these hydrogels. Importantly, this work provides the first set of standards and guidelines for the comparative analysis mucus – a material essential to all animals – in order to extract an understanding of function based on composition.

Comment 2: The authors are claiming in the abstract that they have identified new proteins in the hydrogel what they called CAMPs, without any further explanation. It is not obvious from Figure 3 that all these CAMPs contain signal peptide that allows them to be secreted. If I understand it correctly from figure 2, there was no proteomic identification of these molecules, either. In addition, since the authors claims that the mucins from the snail are quite small, it would have been nice to see SDS-PAGE stained with for instance PAS of their isolated fraction, in order to appreciate the difference between snail and mammalian mucins, of course it would be possible

to perform proteomic also on individual bands. It would also be suggested that the authors are providing additional evidence of the size of the CAMPs and the sequence using for example pcr to determine the transcript of the proposed mucins.

Response: We thank the Referee for their suggestions on how to more effectively characterize CAMP proteins. We have carried out additional analyses to verify the weight of the CAMPs, such that their molecular weights are now confirmed from two complementary datasets – RNA transcriptomics and SDS-PAGE with proteomic mass spectrometry.

While we appreciate the Referee's suggestion that we use PCR to confirm the molecular weights of our CAMPs, we determined that PCR would not achieve the goal and the reasons why are as follows: we felt that PCR would only provide data that we already possessed – weights of the transcripts, which we acquired from RNA sequencing. So, validation of transcripts with PCR would be redundant, as the PCR primers are designed using the sequences determined by RNAseq. In other words, the results of PCR studies would be determined before the experiments are conducted because the primers cannot be designed independent of the RNAseq data, which was first used to determine the sizes of the CAMPs. As such, PCR would not provide complementary validation of CAMPs' sizes.

So, to provide complementary validation of the CAMP sizes, we followed the Referee's other suggestion of using SDS-PAGE. We have optimized SDS-PAGE conditions, and run gels of all three mucus samples using Coomassie, silver, and PAS staining, because "*SDS-PAGE stained with for instance PAS*" and "*proteomic also on individual bands*", were suggested by Referee 2 to verify our "*claims that the mucins from the snail are quite small*" (Referee 2, Comment 2). To do this experiment, the gels were sliced into sections according to molecular weight, and these sections were subjected to proteomic mass spectrometry. We were able to identify the CAMP proteins by mass spectrometry and assign them to different sections of the gels, and, in turn, assign them to molecular weight ranges. The CAMPs were found in the expected molecular weight ranges consistent with RNAseq data. In summary, the sizes of the CAMPs have been determined from the transcriptomes and, separately, from gel electrophoresis/mass spectrometry analysis. These new experiments are described in new text in the manuscript, a description of the experiments in the methods section, and 8 new figures in the Supporting Information, all of which are described in detail below.

For our new SDS page/mass spectrometry analysis, mucus proteins within the gels were sliced into sections of a given molecular weight range, and these bands were subjected to proteomic mass spectrometry. Proteins were separated into bands of three molecular weight ranges: low (<40 kDa), medium (40 – 150 kDa), or high (150+ kDa). We detected 53 of our 71 assigned proteins (75%) in SDS-PAGE proteomic analysis across the three snail mucus samples. We have added the following sentences to the end of the section "**Identification and sequence alignment of snail mucus proteins**"

"To further validate the molecular masses of the identified proteins, SDS-PAGE was conducted on each of the three mucuses to resolve their polypeptides by size. Lanes were divided into three molecular mass regions and excised for proteomic analysis (Supplementary Figures 9 – 11). 53 of the 71 assigned proteins were identified in the gels (Supplementary Figures 12 – 14), and the bands where the proteins were found were consistent with the masses predicted by transcriptomic analysis."

With regards to the CAMPs, we detect CAMP2 and CAMP3 in the adhesive and protective mucus, consistent with our previous findings, though we did not detect CAMP1 in the mass spectra of the gels. In this new analysis, we detect 53 of the 71 assigned proteins (~75%), and detecting 2 of the 3 CAMP proteins is consistent with the overall statistics. This may be the result

of sampling conditions of the snails affecting the expression of CAMP genes and mucus secretion, as mucus is prone to batch-to-batch variation. Importantly, we only detect CAMP2 and CAMP3 in the low molecular weight slices, consistent with the weights of ~48 and 10 kDa predicted from transcriptomic analysis were correct. We have added the following text to the section on CAMPs to reflect our findings:

“Proteomic analysis on SDS-PAGE-separated mucus proteins identified CAMP2 and CAMP3 only in the low molecular mass (< 40 kDa) bands, consistent with the putative transcript sizes determined by RNAseq (Supplementary Figure 15). Interestingly, CAMP1 was not detected in the SDS-PAGE, the reason for which requires further analysis.”

We utilized mass spectrometry to map the glycosylation of CAMP proteins from purified mucus (independent of the SDS-PAGE analysis described above) by identifying the O-glycans attached to CAMP1, CAMP2, and CAMP3. We were able to identify specific glycans in the MS/MS and were able to map these glycans to exact amino acid positions in the protein sequences. We have added the following sentence to the section on CAMPs:

“Glycan site-mapping studies using tandem mass spectrometry on trypsin-digested CAMP peptides revealed that CAMPs are modified with several O-glycans (Supplementary Table 6). CAMP1 contained a FucHex₂HexNAc modification. CAMP 2 was found to have several O-glycans, including FucHex₂HexNAc, (Me)Hex₂HexNAc, HexNAc₂, and HexNAc. CAMP3 is modified with (Me₂)Hex₂HexNAc and Hex₂HexNAc”

To describe the new experiments conducted, we have added the following to the experimental section of the manuscript:

“**SDS-PAGE.** Mucus samples were collected from *C. aspersum* snails in January 2023 in the same manner as described previously. Purified mucus proteins were solubilized in 1 % (v/v) Tween solution and mixed with 2X SDS loading buffer (Quality Biological, 351-082,661). Samples were vortexed and incubated at 95 °C for 15 min before loading in triplicate on a 4 – 20% gradient gel. Electrophoresis was conducted at 150 V for 54 min. Following electrophoresis, gels were divided into three identical sections and each was subjected to Coomassie, Silver, or PAS staining, respectively (Supplementary Figures 9 – 11). Coomassie gel lanes were divided into three slices to obtain molecular mass ranges of < 40 kDa, 40 – 150 kDa, and 150+ kDa. Slices were stored on ice packs prior to proteomic analysis.”

We have added the following detailed procedures to the general methods section of the Supporting Information:

“SDS-PAGE

Lyophilized snail mucus protein samples were suspended in 25 µL of ultrapure water and mixed with an equal volume of 2X SDS loading buffer (Quality Biological, 351-082,661) with 5% (v/v) 2-mercaptoethanol (VWR, M31) and 1 % (v/v) Tween added. Samples were vortexed and incubated at 95 °C for 15 min using a VWR Mini Block Heater (10153-318). 10 µL of reduced protein samples were loaded in triplicate, alongside a Chameleon Duo Pre-Stained protein ladder (LI-COR, 928-60000) onto a 15-well 4 – 20 % Tris-glycine precast gradient gel (BioRad, 4561096) in a BioRad Mini-Protean Tetra system with PowerPac Basic. Gels were electrophoresed at 150 V for 54 min. Following electrophoresis, gels were rinsed with deionized water and each gel was divided into 3 pieces for Coomassie, silver, and PAS staining, respectively. From each gel, a control band from lanes that did not contain protein was sliced to verify gels were not contaminated.

For Coomassie protein staining, gels were fixed in 50 % (v/v) methanol, 10 % (v/v) acetic acid in water solution for 15 min. Gels were then stained in the same solution containing 1% (w/v) Coomassie Brilliant Blue G250 using microwave irradiation. Gels were destained by alternating washes with 50% methanol, 10% acetic acid solution, and ultrapure water.

For silver staining, gels were stained using the Pierce Silver Staining for Mass Spectrometry kit (Thermo Scientific, 24600) according to the manufacturer's instructions.

For PAS staining, gels were incubated in 1 % (w/v) periodic acid solution in the dark for 5 min with occasional shaking. Gels were rinsed with water and incubated in 0.5 % (w/v) sodium metabisulfite solution for 5 min. Gels were rinsed with water and incubated in Schiff's reagent for 5 min in the dark with occasional shaking. Gels were rinsed with water and incubated in 0.5 % (w/v) sodium metabisulfite solution for 5 min. Gels were destained by alternating washes with 50% methanol, 10% acetic acid solution, and ultrapure water.

All gels were imaged simultaneously on an Aversham Imager 600 gel imager (GE) using colorimetric transillumination. Following imaging, Coomassie-stained gel slices were excised using a scalpel. Slices were divided into molecular mass ranges of < 40 kDa, 40 – 150 kDa, and 150+ kDa. Gel slices were stored on ice packs for transport prior to proteomic analysis.

Proteomic Mass Spectrometry of SDS-PAGE Gel Bands

Gel bands were reduced with DTT, alkylated with iodoacetic acid, and digested with trypsin. Extracted peptides were re-solubilized in 0.1% aqueous formic acid and loaded onto a Thermo Acclaim Pepmap (Thermo, 75uM ID X 2cm C18 3uM beads) precolumn and then onto an Acclaim Pepmap Easyspray (Thermo, 75uM X 15cm with 2uM C18 beads) analytical column. Separation was conducted using a Dionex Ultimate 3000 uHPLC at 250 nl/min with a gradient of 2-35% organic (0.1% formic acid in acetonitrile) over 2 hours. Peptides were analyzed using a Thermo Orbitrap Fusion mass spectrometer operating at 120,000 resolution (Full Width at Half Maximum in MS1) with HCD sequencing (15,000 resolution) at top speed for all peptides with a charge of 2+ or greater."

We have also added 8 figures to the Supporting Information describing the results of the SDS-PAGE and the proteomic experiments. Three figures (**Figures S9 – S11**) show the SDS-PAGE gels of adhesive, lubricating, and protective snail mucus. For each mucus sample, proteins were loaded in triplicate, so that each sample could be independently subjected to Coomassie, Silver, and PAS staining, and the results of these experiments are shown below:

Figure S9. SDS-PAGE of adhesive snail mucus using a 15-well 4 – 20 % Tris-glycine precast gradient gel (BioRad, 4561096). Samples (M lanes) were run in triplicate alongside a Chameleon Duo Pre-Stained protein ladder (LI-COR, 928-60000; L lane) and separately stained with Coomassie (left), Silver (middle), and PAS (right). Coomassie-stained M lanes containing purified mucus proteins were sliced into low (< 40 kDa), medium (40 – 150 kDa), and high (150+ kDa) slices, as indicated by red dashed lines, and these slices were used for proteomic analysis.

Figure S10. SDS-PAGE of lubricating snail mucus using a 15-well 4 – 20 % Tris-glycine precast gradient gel (BioRad, 4561096). Samples (M lanes) were run in triplicate alongside a Chameleon Duo Pre-Stained protein ladder (LI-COR, 928-60000; L lane) and separately stained with Coomassie (left), Silver (middle), and PAS (right). Coomassie-stained M lanes containing purified mucus proteins were sliced into low (< 40 kDa), medium (40 – 150 kDa), and high (150+ kDa) slices, as indicated by red dashed lines, and these slices were used for proteomic analysis.

Figure S11. SDS-PAGE of protective snail mucus using a 15-well 4 – 20 % Tris-glycine precast gradient gel (BioRad, 4561096). Samples (M lanes) were run in triplicate alongside a Chameleon Duo Pre-Stained protein ladder (LI-COR, 928-60000; L lane) and separately stained with Coomassie (left), Silver (middle), and PAS (right). Coomassie-stained M lanes containing purified mucus proteins were sliced into low (< 40 kDa), medium (40 – 150 kDa), and high (150+ kDa) slices, as indicated by red dashed lines, and these slices were used for proteomic analysis.

Slices from the Coomassie-stained lanes were subjected proteomic analysis on individual bands. Slices from lanes of the gel that did not contain protein were taken and analyzed as negative controls to ensure detected signals are not from contamination. The results of this are shown in **Figures S12 – S14** that have been added to the Supporting Information and are shown below.

Figure S12. Proteomic abundance of SDS-PAGE-resolved adhesive (Adh) mucus proteins. Control refers to a gel slice that did not contain protein. Low, Med, and High refers to gel slices that contained proteins of molecular mass < 40 kDa, 40 – 150 kDa, and 150+ kDa, respectively.

Figure S13. Proteomic abundance of SDS-PAGE-resolved lubricating (Lub) mucus proteins. Control refers to a gel slice that did not contain protein. Low, Med, and High refers to gel slices that contained proteins of molecular mass < 40 kDa, 40 – 150 kDa, and 150+ kDa, respectively.

Figure S14. Proteomic abundance of SDS-PAGE-resolved protective (Prot) mucus proteins. Control refers to a gel slice that did not contain protein. Low, Med, and High refers to gel slices that contained proteins of molecular mass < 40 kDa, 40 – 150 kDa, and 150+ kDa, respectively.

We also included **Figure S15**, shown below, which shows CAMP2 and CAMP3 protein abundances across all of the SDS-PAGE bands:

Figure S15. Proteomic abundance of CAMP2 and CAMP3 across the three purified snail mucus. Control refers to a gel slice that did not contain protein. Adh: adhesive snail mucus; Lub: lubricating snail mucus; Prot: protective snail mucus. Low, Med, and High refers to molecular mass of < 40 kDa, 40 – 150 kDa, and 150+ kDa, respectively.

These results show that CAMP2 and CAMP3 are found in the low molecular weight (<40 kDa) bands of SDS-PAGE gels. Additionally, these results are consistent with our prior quantification, which showed that CAMP2 is found in all three mucus samples, while CAMP3 was found in the protective mucus.

To identify glycans attached to CAMPs, we employed LC-MS/MS on all three mucus samples after subjecting the mucuses to tryptic digestion to generate mucus glycopeptides. We have added **Table S6** to the Supporting Information that displays the CAMP glycopeptides, which glycan modifications they contain, and which amino acid sites these glycans are mapped to. This table is shown below:

Protein	Peptide	Glycan	Position
CAMP1	R.ALFRS[+673.24293]E.I	HexNAc(1)Hex(2)Fuc(1)	221
	C.GVFYLQT[+527.18502][+14.01570]DR.T	HexNAc(1)Hex(2) 14.0157	29
	R.ET[+673.24293]VVEVR.F	HexNAc(1)Hex(2)Fuc(1)	312
CAMP2	A.T[+406.15875]VGDIVK.F	HexNAc(2)	199
	L.QT[+203.07937]DR.T	HexNAc(1)	34
	V.LGS[+203.07937]M[+15.99492]K.E	HexNAc(1)	13
CAMP3	K.AVRDS[+527.18502]QGK.F	HexNAc(1)Hex(2)	74
	R.DS[+527.18502][+28.03140]QGK.F	HexNAc(1)Hex(2) 28.0314	77

Table S6. Glycopeptidomic analysis of CAMPs.

Comment 3: The term “mucomics” is not very well defined, but to my understanding it would include identification of mucins in the snails mucus. From the dendrogram in figure 2 only 9 mucins were identified by proteomics among the 50+ putative mucins in the samples. Strangely enough the authors claims that they only found 2 clusters of mucins, but the cluster in black on the left hand upper corner also contains “mucins”.

Previous Response: The Referee misunderstands what is meant by ‘mucomics’, even though we had clearly defined this term in the original submission. ‘Mucomics’ means characterization of heterogeneous *mucus* hydrogels through an integrative -omics style approach, rather than investigating one specific protein. This is clearly stated in the last paragraph in the introduction:

“Despite these efforts, it remains unclear how differences in protein structure, ion concentration, glycosylation, and other factors operate synergistically to account for the substantial diversity in mucus material properties.²¹ Here, we apply a systematic comparative analysis of adhesive,²⁰ lubricating,³ and protective²² mucus isolated from *C. aspersum*, which are named in accordance with the materials’ ecological function (Figure 1a). Transcriptomic and proteomic sequencing identified the proteins expressed in each mucus and their abundances. Glycomic mass spectrometry was then employed to identify the structures of the glycans decorating these proteins. Elemental analysis through scanning electron microscopy (SEM) coupled with energy dispersive X-ray spectroscopy (EDX) measured concentrations of various ions in the materials. Atomic force spectroscopy quantified the mechanical properties (elastic modulus, E , and work of adhesion, W) of the three samples. Comparison of these datasets reveal how *C. aspersum* exploit differential protein expression — including a series of previously uncharacterized proteins — glycosylation, and ion concentration are used to explain how these hydrogels behave as adhesives, lubricants, or protective barriers²³ (Figure 1b).”

We acknowledge that we do not use the term “mucomics” in this paragraph, which likely led to the Referee’s confusion, so the paragraph has been edited to now read:

“Despite these efforts, it remains unclear how differences in protein structure, ion concentration, glycosylation, and other factors operate synergistically to account for the substantial diversity in mucus material properties.²³ Here, we apply a systematic comparative mucomics analysis — defined as the combination of genetic, chemical, and material studies to understand the structure-function relationships of mucus — of adhesive,²² lubricating,³ and protective²⁴ mucus isolated from *C. aspersum*, which are named in accordance with the materials’ ecological function (Figure 1a). Transcriptomic and proteomic sequencing identified the proteins expressed in each mucus and their abundances. Glycomic mass spectrometry was then employed to identify the structures of the glycans decorating these proteins. Elemental analysis through scanning electron microscopy (SEM) coupled with energy dispersive X-ray spectroscopy (EDX) measured concentrations of various ions in the materials. Atomic force spectroscopy quantified the mechanical properties (elastic modulus, E , and work of adhesion, W) of the three samples. Comparison of these datasets reveal how *C. aspersum* exploit differential protein expression — including a series of previously uncharacterized proteins — glycosylation, and ion concentration are used to explain how these hydrogels behave as adhesives, lubricants, or protective barriers²⁵ (Figure 1b).”

We also hope to clear up the Referee’s misunderstanding regarding the dendrogram. The black cluster in the upper left corner does not contain any of our identified proteins, and are not

described in detail in the results nor discussion sections, but kept for completeness of the data. This labelling is explained in the caption of Figure 2:

“Figure 2. Dendrogram of snail mucus proteins based on sequence similarity. Clusters are colored according to protein function. The “Unclustered” (black) classification indicates a clade that had no discernible function or only contained reference proteins. Proteins identified in this study are labelled with circles. Circle color indicates the protein was found in adhesive (red), lubricating (blue), or protective (green) mucus. An outgroup, three *Mus musculus* proteins (Pikachurin1, Pikachurin2, Pikachurin3), is marked with a bracket. Dendrogram with branch lengths and included species is shown in Supplementary Figure 8.”

While we agree mucins are an important component of mucus hydrogels, mucuses are complex networks of proteins from many families, and the properties of these gels arise from the combined action of all the components, and, as such, the identification of all proteins is highly valuable. Additionally, as pointed out by the Referee (*Referee 2, Comment 5*) and Referee 3 (*Referee 3, Comment 11*), mucins are typically challenging to identify via shotgun proteomics because of their dense glycosylation. We now acknowledge this issue in the first paragraph of the section, ‘**Characterization of adhesive, lubricating, and protective *C. aspersum* mucus proteins**’, by adding the sentence:

“It should be noted that mucins are challenging to identify via shotgun proteomics, because their dense glycosylation limits enzymatic digestion,³⁰ or via transcriptomics because of their tandem repeats³¹.”

Citing the following articles:

30. Nicholas, B. *et al.* Shotgun proteomic analysis of human-induced sputum. *Proteomics* **6**, 4390-4401 (2006).

31 Tørresen, O. K. *et al.* Tandem repeats lead to sequence assembly errors and impose multi-level challenges for genome and protein databases. *Nucleic Acids Research* **47**, 10994-11006, doi:10.1093/nar/gkz841 (2019).

Comment 4: This reviewer fail to find the CAMPs in the dendrogram.

Previous Response: We thank the Referee for their attention to detail and correctly pointing out the missing B–C mucins. We realized there was an oversight on our part, and that an early draft of the dendrogram was included where these proteins were erroneously named “PutativeMucin” and has been changed to “CAMPs” (which we have used instead of BCMucin in response to the Referee’s concerns). As such, Figure 2 has been replaced, and the CAMPs now appear in the dendrogram. The original version of the dendrogram is provided below:

And the new version has been provided below with an arrow to indicate where the CAMPs appear:

Comment 5: Few of the “mucins” detected by proteomics are indeed recognized as mucins when they are found in other species. Very few of the proteins named as MUC was identified by proteomics Mucin type molecules are notoriously difficult to detect in proteomics, indicating that there may be technical difficulties to identify what are the “mucins” that are actually present in the snail secretion.

Previous Response: The Referee points out an important challenge in mucin analysis. We agree that mucins are extremely difficult to identify with proteomics alone, and went to great lengths to ensure we did not incorrectly identify proteins as mucins. In our work, we did not find proteins that could be definitively identified as mucins. Rather, we found that several proteins are mucin-like. As mentioned in an earlier comment (Referee 2, Comment 3), while mucins are an important

component of mucus hydrogels, they are complex networks of proteins from many families. Again, we have added the following sentence to address the challenge of identifying mucins:

“It should be noted that mucins are challenging to identify via shotgun proteomics, because their dense glycosylation limits enzymatic digestion³⁰, or via transcriptomics because of their tandem repeats³¹.”

Furthermore, some of the proteins identified are mucin-like, and we explain this in the section, **Characterization of adhesive, lubricating, and protective *C. aspersum* mucus proteins**. We believe this section, as is, effectively demonstrates this idea and explains similarity between our identified proteins to know mucins:

“A jagged-1-like protein (Jagged1), which is involved in extracellular signaling pathways,²⁷ had sequence similarity to MUC2 from *Pygocentrus nattereri* (Red-bellied piranha). A spondin-like protein (Spondin1), which mediates cell-extracellular matrix interactions,²⁸ also displayed similarity to MUC2, MUC5AC, MUC12, MUC16, and MUC19 from mollusks and other marine life. Spondin1’s sequence features several short regions that are either Ser- or Thr-rich. Curiously, these regions alternate between being Ser-rich and Thr-rich, meaning each region only incorporates one of these two amino acids. This protein is also 12% Cys by composition, which is more than five times greater than the natural abundance of Cys in invertebrate proteins,²⁹ suggesting it has a propensity to form disulfide bridges. It is likely Spondin1 is a *C. aspersum* mucin because it contains repeating O-glycosylated regions, similar to mucin ‘B’ domains, and Cys-rich regions which can multimerize the protein by functioning like mucin ‘A’ domains.”

It is possible other proteins, such as the “Snail” and “Novel” proteins, may be similar to mucins, however sequence databases on mollusk mucus proteins are limited, which precludes assigning definitively function to these proteins. Furthermore, as we noted earlier in this comment, it would be premature to label any protein a mucin without glycomic analysis of the individual protein. In our analysis, we went to great lengths to ensure we did not call anything a mucin, but rather ‘mucin-like’, suggesting further characterization is needed before a definitive assignment can be made. We hope the Referee will agree that this work reinforces the need for further characterization of mucus from invertebrates and mollusks, which will allow for more robust bioinformatic analysis.

Comment 6: In general, there are very few of the proteins identified by transcriptomics that are detected in the mucus by proteomics. This indicates to me that there are technical issues concerning the “mucomics”.

Previous Response: We respectfully disagree with the Referee’s comment regarding the number of proteins identified via proteomic analysis. In this work, we conducted purification of the mucus hydrogel to isolate only the proteins that physically comprise the mucus hydrogel. Additionally, transcriptomics will generate a comparatively massive dataset (179,552 genes) that captures every gene transcribed by the cells, where only a small subset of those genes will actually incorporate into the mucus hydrogel. As a result, we detect with proteomics 71 unique mucus proteins with high statistical confidence according to analytical standards set by the proteomics community, as specified in the section, **Identification and sequence alignment of snail mucus proteins**:

"From the 179,552 transcripts, 71 provided coding sequences for proteins based on the standard criteria of having a minimum of two identified peptides and a false discovery rate of less than 1.0 %.^{18,28}"

It is inappropriate to correlate the size of the transcriptomic database (which contains 6 biological replicates and can be very large) with the number of proteins that should be definitively identified as incorporated into the mucus. While there may be proteins that were disregarded because of the stringent criteria we set for definitive identification, we chose to publish only the high-quality assignments. As the raw transcriptomic and proteomic data was uploaded to Genbank and PRIDE, respectively, other scientists are free to investigate the entire dataset, including lower quality proteins, which is exactly what this manuscript is meant to encourage. It should also be noted that it is a subjective opinion as to whether we identified 'few' or many proteins, but we are unaware of any other report that identifies as many proteins that are integrated within the mucus gels.

Comment 7: The authors may also justify why they need to introduce another -omics, and what this new term actually means. Maybe something like system mucus biology would make more sense?

Previous Response: The term mucomics encompasses the integrative nature of the approach towards, combining analysis of RNAs, proteins, glycans, and ions with materials characterization to understand mucus. The term, like other 'omics' studies, emphasizes that connections between datasets are the keys to thoroughly understanding mucus function. Finally, introducing another 'omics' is justified because mucus is unique in that understanding mucus requires glycomics, proteomics, transcriptomics, and the other methods described above, and no other 'omics' studies require this particular set of analyses. In addition, introducing another 'omics' also brings attention to this understudied, but essential and ubiquitous, class of materials — mucuses. Nevertheless, to address the Referee's concern, we have edited the abstract to better define mucomics. The sentence:

"Here, we characterize mucus proteins, glycosylation, ion content, and mechanical properties to understand structure-function relationships."

Now reads:

"Here, we characterize mucus proteins, glycosylation, ion content, and mechanical properties to understand structure-function relationships through an integrative "mucomics" approach."

Additionally, as described in the response to *Referee 2, Comment 3*, we now define mucomics in the manuscript "as the combination of genetic, chemical, and material studies to understand the structure-function relationships of mucus."

Comment 8: Glycomics data: For N-linked the authors reports that some structures contain only NeuAc while others only NeuGc. This is a strange result that is left without any explanation.

Previous Response: We thank the Referee for pointing out this detail. We have edited the discussion to propose a possible reason for Neu5Gc being present only in the adhesive mucus, specifically that the pedal tissue cannot biosynthesize Neu5Gc from Neu5Ac (*Appl. Microbiol.*

Biotechnol., **2012**, *94*, 887-905). We now state: “We thank the Referee for pointing out this detail. We have edited the discussion to propose a possible reason for Neu5Gc being present only in the adhesive mucus, specifically that the dorsal tissue cannot biosynthesize Neu5Gc from Neu5Ac (*Appl. Microbiol. Biotechnol.*, **2012**, *94*, 887-905).”

Comment 9: Also, there are no attempt to verify the glycans for instance by exoglycosidases digestion or MS/MS.

Response: In response to this comment, we conducted new experiments to acquire additional verification of our glycomic data. We performed tandem mass spectrometry on trypsin-digested glycopeptides to provide further support of the monosaccharide assignments, including confirming assignments of sialic acids. This technique allows us to detect fragmented peptides and observe ion signatures to confirm the glycan assignments, and also to confirm their presence in our protein samples with high accuracy. In response to these new data, we have included changes to the manuscript text, descriptions of these experiments in the methods sections, 6 new figures in the Supporting Information, and 3 tables identifying all glycans detected by this method. We have edited the text to reflect the new data by adding the following paragraph to the end of the section “**Glycomic analysis of *C. aspersum* mucus**”:

“The purified mucuses were subjected to tryptic digestion and tandem mass spectrometry to identify mucus glycopeptides and verify glycan assignments (Supplementary Tables 9 – 11).⁶⁶ Characteristic HexNAc, Neu5Ac, and Neu5Gc m/z peaks confirmed the presence of these monosaccharides in all three mucus samples (Supplementary Figures 22 – 24). To further validate the presence of sialic acids on mucus proteins, fragmentation patterns of Neu5Ac-containing glycopeptides were detected through the MS/MS, confirming that snail mucus proteins are modified with sialic acids (Supplementary Figures 25 – 27). While low-abundance oxonium ions for Neu5Gc were detected in the tandem mass spectra, MS/MS analysis could not detect Neu5Gc-modified glycopeptides.”

We have also added the following paragraphs to the methods sections of our manuscript and the supporting information:

“Glycoproteomic Tandem Mass Spectrometry:

Purified mucin powders were suspended in 100 μ L of 50 mM ammonium bicarbonate. To this, 100 μ L of 25 mM DTT was added. Samples were vortexed and incubated at 45°C for 45 minutes. Following incubation, samples were allowed to cool to room temperature, and 100 μ L of 90 mM iodoacetamide (IAA) was added. Samples were vortexed and incubated at room temperature in the dark for 20 minutes. Following incubation, samples were cleaned with 3 kDa MWCO filters (Millipore Amicon Ultra, UFC500396). Prior to loading samples, the filters were washed twice with 400 μ L of 50 mM ammonium bicarbonate. Filters were spun for 10 minutes at 14000 rpm. Following washing, samples were loaded onto the filters and spun at 14000 rpm for 25 minutes. 400 μ L of 50 mM ammonium bicarbonate was added, and samples were spun once more. This process was repeated one more time. The desalted protein sample was removed from the filter by inverting it into a clean tube and centrifuging for 5 minutes. The filters were rinsed with 50 μ L of 50 mM ammonium bicarbonate, and the samples were inverted once more and centrifuged. The total volume was then brought to 100 μ L. A 5 μ L aliquot was digested with 1 μ g of trypsin (Promega) overnight at 37°C. Trypsin was then terminated by heating samples to 100°C for 5 minutes. Samples were then passed through a 0.2 μ m filter. Samples were diluted to 30 μ L in 0.1% formic acid.

Samples were analyzed using a Thermo Fisher Eclipse Tribrid mass spectrometer equipped with a nano electrospray source and coupled to an Ultimate 3000 RSLCnano liquid chromatography system. Samples were analyzed using a 180-minute gradient. A prepacked nano-LC column of 15 cm length and 75 μm internal diameter, filled with 3 μm C18 material was used. The precursor ion scan was acquired at 120,000 resolution in the Orbitrap, and precursors with a time frame of 3 seconds were selected for MS/MS fragmentation in the Orbitrap at 15,000 resolution. Monoisotopic precursor selection was selected and the threshold for MS/MS triggering was 1000 counts. MS/MS fragmentation was done using stepped higher energy collision induced dissociation (HCD) product triggered collision induced dissociation (CID) (HCDpdCID). Precursors with an unknown charge state or charge state of +1 were excluded, and samples were run in positive ion mode.

LC-MS/MS spectra were searched against the FASTA sequences of the Helicidae genome obtained from Uniprot, as well as the protein sequences obtained from proteomics experiments and RNAseq experiments outlined in this paper. Byonic software (Ver 5.0.12) was used for data analysis. Oxidation of methionine, deamidation of asparagine and glutamine were searched as variable modifications and carbamidomethylation of cystine was searched as a fixed modification. A Byonic *N*-glycan database of insect and plant glycans as well as a tailored glycan list developed based on the glycomics results were searched as variable modification for *N*-glycans. A Byonic *O*-glycan database of 9 common *O*-glycans as well as a tailored glycan list developed based on our glycomics results were searched as variable modification for *O*-glycans. The data was then filtered based on a $|\log \text{prob}|$ value equal to or greater than 3, and a Delta Mod Score equal to or greater than 50. Matches were manually verified by confirming presence of oxonium ions and expected neutral loss patterns.“

To provide support for our glycomic assignments, we have added **Tables S9 – S11** showing glycans determined by tandem mass spectrometry. Many of these glycans are consistent with those detected in the initial glycomic analysis, verifying our structural assignments. As an example, **Table S9**, which shows glycans in the adhesive mucus determined by MS/MS, is shown below:

Mucus	Glycosylation Type	m/z	Glycan
Adhesive	O	203.07937	HexNAc(1)
Adhesive	O	365.1322	HexNAc(1)Hex(1)
Adhesive	O	406.15875	HexNAc(2)
Adhesive	O	527.18502	HexNAc(1)Hex(2)
Adhesive	O	541.20072	(Me)HexNAc(1)Hex(2)
Adhesive	O	568.21157	HexNAc(2)Hex(1)
Adhesive	O	656.22761	HexNAc(1)Hex(1)NeuAc(1)
Adhesive	O	673.24293	HexNAc(1)Hex(2)Fuc(1)
Adhesive	O	703.25354	(Me)HexNAc(1)Hex(3)
Adhesive	N	203.07937	HexNAc(1)
Adhesive	N	349.13728	HexNAc(1)Fuc(1)
Adhesive	N	406.15875	HexNAc(2)
Adhesive	N	495.19519	HexNAc(1)Fuc(2)
Adhesive	N	552.21665	HexNAc(2)Fuc(1)
Adhesive	N	568.21157	HexNAc(2)Hex(1)
Adhesive	N	698.27456	HexNAc(2)Fuc(2)
Adhesive	N	714.26948	HexNAc(2)Hex(1)Fuc(1)
Adhesive	N	730.26439	HexNAc(2)Hex(2)
Adhesive	N	860.32739	HexNAc(2)Hex(1)Fuc(2)
Adhesive	N	876.3223	HexNAc(2)Hex(2)Fuc(1)
Adhesive	N	892.31722	HexNAc(2)Hex(3)
Adhesive	N	906.33292	(Me)HexNAc(2)Hex(3)
Adhesive	N	1022.38021	HexNAc(2)Hex(2)Fuc(2)
Adhesive	N	1024.35947	HexNAc(2)Hex(3)Pent(1)
Adhesive	N	1038.37512	HexNAc(2)Hex(3)Fuc(1)
Adhesive	N	1052.39087	(Me2)HexNAc(2)Hex(3)Pent(1)
Adhesive	N	1054.37004	HexNAc(2)Hex(4)
Adhesive	N	1065.38602	HexNAc(3)Hex(2)Pent(1)
Adhesive	N	1095.39659	HexNAc(3)Hex(3)
Adhesive	N	1170.41738	HexNAc(2)Hex(3)Fuc(1)Pent(1)
Adhesive	N	1184.43303	HexNAc(2)Hex(3)Fuc(2)
Adhesive	N	1227.43885	HexNAc(3)Hex(3)Pent(1)
Adhesive	N	1241.4545	HexNAc(3)Hex(3)Fuc(1)
Adhesive	N	1257.44941	HexNAc(3)Hex(4)
Adhesive	N	1298.47596	HexNAc(4)Hex(3)
Adhesive	N	1346.48586	HexNAc(2)Hex(4)Fuc(2)
Adhesive	N	1362.48077	HexNAc(2)Hex(5)Fuc(1)
Adhesive	N	1378.47569	HexNAc(2)Hex(6)
Adhesive	N	1387.51241	HexNAc(3)Hex(3)Fuc(2)
Adhesive	N	1403.50732	HexNAc(3)Hex(4)Fuc(1)
Adhesive	N	1430.51822	HexNAc(4)Hex(3)Pent(1)
Adhesive	N	1519.55466	HexNAc(3)Hex(3)Fuc(2)Pent(1)
Adhesive	N	1540.52851	HexNAc(2)Hex(7)
Adhesive	N	1565.56014	HexNAc(3)Hex(5)Fuc(1)
Adhesive	N	1606.58669	HexNAc(4)Hex(4)Fuc(1)
Adhesive	N	1670.5915	HexNAc(2)Hex(6)Fuc(2)
Adhesive	N	1702.58133	HexNAc(2)Hex(8)
Adhesive	N	1752.6446	HexNAc(4)Hex(4)Fuc(2)
Adhesive	N	1850.69261	HexNAc(6)Hex(3)Fuc(1)
Adhesive	N	1900.68178	HexNAc(4)Hex(5)Fuc(1)Pent(1)
Adhesive	N	1914.69743	HexNAc(4)Hex(5)Fuc(2)
Adhesive	N	2336.85109	HexNAc(6)Hex(6)Fuc(1)
Adhesive	N	3432.24767	HexNAc(9)Hex(9)Fuc(1)

Table S9. Glycans extracted from adhesive *C. aspersum* snail mucus proteins that were detected via glycoproteomic tandem mass spectrometry analysis.

We were able to verify through LC-MS/MS the compositions of our assigned glycans. These assignments include less-common monosaccharides, such as sialylated sugars, methylated sugars, and pentoses. Similar tables detailing glycans identified in this manner from snail lubricating (**Table S10**) and protective (**Table S11**) are included in the supporting information.

Additionally, we have added **Figures S22 – S24** showing LC-MS/MS chromatograms of snail mucus glycopeptides. These figures show chromatograms that verify the presence of HexNAc, Neu5Ac, and Neu5Gc in the mucus protein samples by detecting their respective m/z signals in the adhesive, lubricating, and protective mucus. One example of these figures, **Figure S23**, demonstrating the presence of Neu5Ac in all three mucus secretions is shown below:

Figure S23. Extracted ion chromatograms for m/z 292.1026 (Neu5Ac signal). Peaks indicate presence of Neu5Ac at the corresponding time point.

These chromatograms show the traced signal of m/z 292.1026 during the LC-MS/MS analysis, which is characteristic of Neu5Ac. This data confirms that MS/MS is able to detect appreciable levels of Neu5Ac in the adhesive and protective mucus, but the signal is much less abundant in the lubricating mucus. These data are consistent with our previous glycomics results, which show that sialylation is prominent in the adhesive and protective mucus. We have included similar figures for HexNAc and Neu5Gc in the Supporting Information.

We have also included new figures, **Figures S25 – S27**, of MS/MS spectra in the Supporting Information showing Neu5Ac-linked glycopeptides were detected in the adhesive, lubricating, and protective mucus. One example spectrum from the adhesive mucus, **Figure S25**, is shown below, where we show an example MS/MS spectrum of a Neu5Ac-containing glycopeptide from LC-MS/MS of snail adhesive mucus proteins. We detect signals of Neu5Ac and glycans from the digested glycopeptide, providing further support for the glycan assignments, including for sialic acids, which are traditionally challenging to confirm. We included similar spectra for glycopeptides from snail lubricating and protective mucus.

Figure S25. Tandem mass spectrum of Neu5Ac-containing glycopeptide from snail adhesive mucus. Spectrum is taken from LC retention time of 40.81 min.

Comment 10: Elemental analysis. All analysis (Supplementary figure S19-21) contains high amount of aluminum. One can expect that this is a contamination. With this high amount of contamination, the measurement of other metal ion is put in doubt.

Previous Response: We respectfully disagree with the Referee that the aluminum is contamination. We describe in the Methods section how the samples are collected:

“To create the samples, live snails were allowed to crawl on SEM Al pin stubs (Ted Pella, 16144) that were inverted or horizontal, to create samples for adhesive and lubricating mucus, respectively, and back (protective) mucus was scraped onto the stubs, similar to the silicon wafer samples for AFM, and air-dried overnight.”

The SEM/EDX substrates are aluminum pin stubs, thus the high amounts of aluminum are not contamination, but rather expected as the background signal. As shown in the SEM/EDX overlays in the supporting figure below, the signal for elements present in the mucus are most prominent (intensely colored) where the Al signal is minimal (no coloration), as the Al signal only appears where there is no material present, or in other words because only the aluminum stub is being detected. As such, no changes have been made in response to this comment.

Adhesive Mucus

Comment 11: Page 5. What is proteomic abundance?

Previous Response: We thank the Referee for pointing out the confusion regarding protein quantification. To answer the Referee’s question, we have added more detail to the sentence:

“All proteins were quantified based on proteomic abundance (Supplementary Figures 5–7).”

Which now reads:

“All proteins were quantified based on MS/MS proteomic abundance (Supplementary Figures 5–7, Supplementary Table 2).”

Additionally, as addressed earlier (*Referee 1, Comment 9*), we have added a supporting figure (Table S2) including MS/MS quantification.

Comment 12: Page 9 The sentences about L-dopa role in snail has not been substantiated in this report. Much of this is speculation and should be moved to discussion

Previous Response: We respectfully disagree with the Referee that comments about L-Dopa should be removed. The language regarding L-DOPA’s role in snail mucus is intentionally speculative, which is not to be confused with unfounded, because we are suggesting how it could have function in these materials based upon extensive literature precedent. We would not have added this speculation were there no a strong literature precedent, and we make this very clear in our writing. We also feel that, upon discovering these proteins, it would be inappropriate to not make informed suggestions on their possible role in mucus and add context. Nevertheless, to address the Referee’s concerns, we have edited the following sentences as to not imply we are reporting a definitive role of these proteins in the mucus:

“A tyrosinase, Tyrosinase1, was found exclusively in the adhesive mucus, which catalyzes the formation of L-DOPA from tyrosine. As L-DOPA is involved in forming strong glues in mussels, this observation suggests that *C. aspersum* land snails may use a similar adhesive mechanism as marine mollusks, which has not been reported previously.”

Now reads:

““A tyrosinase, Tyrosinase1, was found exclusively in the adhesive mucus, which catalyzes the formation of L-DOPA from tyrosine. As L-DOPA is involved in forming strong glues in mussels, this observation suggests that *C. aspersum* land snails may use a similar adhesive mechanism as marine mollusks.”

REVIEWER COMMENTS

Reviewer #2 (Remarks to the Author):

I have now read the new improved version of the manuscript “Comparative Mucomic Analysis of Three Functionally Distinct Cornuaspersum Secretions” by Cerullo et al. I am very happy that the authors have performed additional analysis as I suggested in my previous review: especially the proteomic/glycoproteomic analysis where they now positively identified the new interesting CAMP-proteins.

The report provides substantial amount of data from transcriptomic, proteomic, glycomic, elemental analysis, electron microscopy and AFM and it is obvious that this approach is providing interesting leads for the authors to follow up.

However, as I already written, the different analytical approaches are only providing leads and the authors are only using literature information to do the connection between structure and function. The authors say themselves in the abstract “Here, we characterize mucus proteins, glycosylation, ion content, and mechanical properties to understand structure-function relationships through an integrative “mucomics” approach”. I think that this statement is misleading and without any verification of structure-function, the manuscript in my opinion is lacking the impact that I believe is justifying a publication in Nature Communication. I acknowledge that verifying all the potential leads is a lot of work, but it would make sense that the authors would at least attempt to verify the function of some of the features described in Figure 1b with the physiological function of the various mucus sample. The author strangely enough does not have any appetite in doing so, despite that this would be the next logical step.

Additional concerns

The authors did indeed perform the SDS-PAGE with subsequent staining and proteomics and highlights a significant piece of information that was not obvious in the previous versions. The proteomic analysis of the SDS-PAGE separated components shows that there is a dominating component in the +260 kDa region (Figure S9-S11), that is stronger staining than any of the “streaky” bands detected by silver staining. Without a definite identification of what this glycoprotein is or glycoproteins are, in my opinion any discussion relating structure with the functions reported would be too much of speculation. I do appreciate that proteomic identification of what looks like a mucin type component is difficult. However, the identity of this component may totally shift the focus of a structure-function relationship that the authors try to address.

It is also a concern that only 71 (only 64 reported in figure 2) out of +200-300 predicted proteins in Figure 2 was identified by proteomics. It raises the concern for how comprehensive the proteomic analysis is, even though the author claim it is an “inappropriate” comparison. I relate back to the authors

own agenda that they want to do structure- function correlation, I think it is a valid point to ask if they have identified the components that are responsible for the macroscopic function of the different mucus samples.

CAMPs: The authors claims that they have confirmed the size of the CAMPs by transcriptomics. The first question I would ask is how they know that the sequences they present are the fully translated sequence, none of them presents an N-terminus with an initiating Methionine. Glycosylation. Please provide spectra so that we clearly can see b/y ions as well as glycorelated fragment. Please explain in the table S6 that +14 and +28 probably is methyl and +16 is oxidation of methionine as you did in the glycomic tables. Did you only detect one glycoform of each peptide? Was the identification of these as stringent as the other proteomic data, or where there different criteria for their identification? Figure 3B, how does the glycosylation reported there correspond to the glycosylation detected in table S6. Was the N-linked glycosylation also confirmed? In table S6 you had 1 O-linked for CAMP1, 5 for CAMP 2 and 2 for CAMP 3. How does this relate to the amount of O-linked and N-linked in figure 3B, or are these figures not relating to the actual analytical data? Also, I don't agree that the proteomic confirms the size of the CAMPs, since mucins are often found to be proteolytically cleaved and smaller fragment can often be detected.

Glycomics: The difference in lubrication/adhesiveness/protection and the length of the glycans is contradictory, since in the literature referred to in this subject actually shows that the glycans shown for the adhesive mucus is identical to the glycans they are referring to that provide the best lubricating molecules known in biological systems. N-Glycolyl neuraminic acid not very well substantiated (Figure S24), all samples appear to have NeuGc looking at the extracted ion chromatogram in S24. No MS/MS confirmation on glycomic or glycoproteomic level. Only evidence is the low intense mass found by glycomic analysis that we are supposed to believe corresponds to a composition containing NeuGc. It is also recommended that the authors stick to the minimum reporting standards for glycomics (MIRAGE) and it is advisable that they submit their data into repositories such as glycopost.

Loading: It is obvious that the samples subjected to SDS-PAGE and proteomic analysis is different, where the protective samples contained more material (Supplementary Figure 9-11) This is despite that the authors says that they loaded equal amount for the SDS-PAGE. This makes figures S12-14 difficult to compare the The authors did not describe in the experimental section if the shot gun proteomics suffered from the same problem, and if this caused a bias when comparing the proteins in the samples. The same for the glycomic analysis (here strangely they refer to the samples as "mucin" samples).

Mucomics: I still don't see the necessity in introducing a -omic, for analysing mucus by various omics method. This sounds more what would be defined as multiomics of mucus or mucus multiomics. Otherwise we would have to define a new omics whenever we are analysing a new type of tissue or cell with a multiomics approach.

REFEREE 2

*Comment 1: I have now read the new improved version of the manuscript “Comparative Mucomics Analysis of Three Functionally Distinct *Cornuaspersum* Secretions” by Cerullo et al. I am very happy that the authors have performed additional analysis as I suggested in my previous review: especially the proteomic/glycoproteomic analysis where they now positively identified the new interesting CAMP-proteins. The report provides substantial amount of data from transcriptomic, proteomic, glycomic, elemental analysis, electron microscopy and AFM and it is obvious that this approach is providing interesting leads for the authors to follow up.*

Response: We thank the Referee for stating that our work is substantial and interesting. We appreciated the Referee’s suggestions, and we are pleased that they communicate their satisfaction with the new experimental data.

Comment 2: However, as I already written, the different analytical approaches are only providing leads and the authors are only using literature information to do the connection between structure and function. The authors say themselves in the abstract “Here, we characterize mucus proteins, glycosylation, ion content, and mechanical properties to understand structure-function relationships through an integrative “mucomics” approach”. I think that this statement is misleading and without any verification of structure-function, the manuscript in my opinion is lacking the impact that I believe is justifying a publication in Nature Communication.

Response: **Regarding structure-function relationships**, we recognize the Referee’s concerns, and we are changing the text to ensure we do not imply an overbroad understanding of these relationships, and in doing so we hope to satisfy the Referee’s concerns. The sentence which read:

“Here, we characterize mucus proteins, glycosylation, ion content, and mechanical properties to understand structure-function relationships through an integrative “mucomics” approach”

Has been changed to:

“Here, we characterize mucus proteins, glycosylation, ion content, and mechanical properties that could be used to provide insight into structure-function relationships through an integrative “mucomics” approach.”

Regarding impact, we disagree with the Referee’s opinion that “the manuscript in my opinion is lacking the impact that I believe is justifying publication in *Nature Communication*[sic] .” Here we have accomplished the following:

- We have reported 71 proteins that are components of the snail mucus gels, including collagen, serpin, and epiphragmin classes, and this dataset contains 18 proteins that have never before been identified.
- We have reported the glycan, ion, and protein compositions, as well as the mechanical properties of snail mucus.
- We have, for the first time, generated *comparable datasets* between lubricating, adhesive, and barrier mucus from the same animal, allowing for the identification of differences between the materials and providing the first data that could be used to build strong hypotheses regarding structure-property relationships.
- Entire books are now being written that are devoted to the wonders of mucus (Wedlich, Susanne. *Slime: A Natural History*. Melville House, 2023.), in which our comparative studies of mucus described in this manuscript are mentioned specifically for its potential to make breakthroughs in understanding the composition, properties, and function of mucus.

At the same time, there are large communities of researchers from industry and academia that are seeking the exact information that the manuscript reports. As already described in the introduction, these communities include:

- The multibillion-dollar snail mucus cosmetics industry, currently estimated at \$3.7 billion annually, and estimated to grow to \$12 billion by 2029.¹ In particular, the industry will be interested in our finding that collagen is prevalent in snail mucus, which has become an extremely sought-after ingredient in cosmetics, nutrition, and dermatology for its uses in skincare products, wound healing, and tissue rejuvenation.²⁻⁷
- Biomedical scientists who are currently exploring snail mucus as drug delivery agents;⁸ antibacterial, antifungal, and anticancer agents;⁹ additives in durable antimicrobial and UV-protective materials in food packaging;¹⁰ formulations for treating ocular diseases;¹¹ promoters for tissue culture;¹² and as surgical adhesives with wound healing capabilities.¹³
- Biologists and biochemists who are interested in the role of mucus in predation and reproduction,¹⁴ maintaining homeostasis under stress,¹⁵ deterring predators,¹⁶ and determining host-microbiome dynamics.¹⁷

Clearly, all of these groups are going to be very interested in the composition of the material that they are so invested in, and in particular the high quality and thorough data presented in the submitted manuscript. We also want to point out that *Nature Communications* has published an article on snail mucus *this year*,¹³ showing that other work in the field, which would benefit greatly from our findings, is of interest to the readers of this journal. Furthermore, this article will completely change how mucus is studied and described. As such, between the large community interest and the data itself, the Referee's assertion that is work is "lacking...impact" is simply not accurate. We believe that this article will be amongst the top viewed and downloaded articles published by *Nature Communications* this year, and we look forward to reaching a particularly broad audience because the article will be open access.

Comment 3: I acknowledge that verifying all the potential leads is a lot of work, but it would make sense that the authors would at least attempt to verify the function of some of the features described in Figure 1b with the physiological function of the various mucus sample. The author

strangely enough does not have any appetite in doing so, despite that this would be the next logical step.

Response: The composition of snail mucus is incredibly complex and comparing the composition of three mucus – as we have already done in this manuscript – is even more so. In this work we have identified the protein, glycan, and ion compositions, as well as the mechanical properties, of three distinct mucus from the snail *C. aspersum*. Our work already exceeds substantially in scope, ambition, and quality any previous study on the composition of animal mucus secretions, and this comprehensive and detailed work, as well as the datasets we provide, will inevitably lead to new insight. In addition, anyone knowledgeable in the art of characterizing novel, multicomponent biomaterials would recognize that characterizing even some of the identified proteins is far outside the scope of a single paper and would be particularly challenging because every measurable material property is certainly the result of synergistic interactions of multiple components. If a requirement for publication were to fully explore each of these leads, the barrier for publication would be so high that no work would ever be published.

Regarding the verification of function of individual proteins and the Referee's comment that "it would make sense that the authors would at least attempt to verify the function of some of the features described in Figure 1b with the physiological function of the various mucus sample", there are two important things to note:

- The first point is that it should be clearly recognized what Figure 1b reports – the figure simply reports what components are present in the three different mucus from the proteomic, glycomic, ionic, and material characterization data and does not describe any functional characterization of the individual components.
- Regarding the second point, where the Referee is requesting that we further characterize the function of the individual materials, we would push back strongly against this. An analogous request would be like asking the authors who first reported the completed human genome to, in the same publication, report the function of each gene and the interactions with other genes – questions which will likely take another century of research to completely resolve. Similarly, although perhaps not to the same scale, the Referee is asking that, in the same publication, we describe all the contents of snail mucus and that we provide a complete description of how all the various components interact to result in 'function'. Certainly, we are continuing to explore these interactions between various components of the mucus, and we hope the Referee would understand why such studies are outside the scope of this already complex manuscript. Moreover, even if all these data could be and were already collected, reporting all these data in a single publication would result in a paper so complex that it would dilute the message and reduce clarity of the manuscript, which we do believe would lessen its impact.

As such, we have not made any changes in response to this comment.

Comment 4: The authors did indeed perform the SDS-PAGE with subsequent staining and proteomics and highlights a significant piece of information that was not obvious in the previous versions. The proteomic analysis of the SDS-PAGE separated components shows that there is a dominating component in the +260 kDa region (Figure S9-S11), that is stronger staining than any of the "streaky" bands detected by silver staining. Without a definite identification of what this glycoprotein is or glycoproteins are, in my opinion any discussion relating structure with the

functions reported would be too much of speculation. I do appreciate that proteomic identification of what looks like a mucin type component is difficult. However, the identity of this component may totally shift the focus of a structure-function relationship that the authors try to address.

Response: We thank the Referee for the great suggestion to conduct SDS-PAGE-proteomic analysis on the 260 kDa band, which revealed prominent glycoproteins in the mucus. To address the Referee's comment, there are two points to be made on our new identification of the proteins in the band at 260 kDa, and how these new findings do not in any way alter the focus or message of the manuscript.

- **Regarding the identification of the components of the 260 kDa band**, we repeated SDS-PAGE on the three mucus and excised this specific ~260 kDa band from all three samples to perform proteomic and glycoproteomic analysis. These analyses identified six proteins and three of these, Novel4, Novel7, and Novel8, were present in all samples of that band that were subjected to the proteomic analysis. In addition, we conducted glycoproteomic site-mapping studies on each of these proteins to identify the diversity and extent of their glycan modifications. We have added a paragraph to the manuscript text to reflect these new findings (see below).
- **Regarding how these new results impact the goals of our work**, it is important to note that even with this new identification, it does not in any way change the focus of this manuscript, which is to report comparable datasets of the protein, glycan, and ion content, and material properties of three different mucus secretions from *C. aspersum*. With these datasets, we and the readers of this manuscript can begin to develop structure-function relationship hypotheses on these complex materials. These hypotheses, backed by our comprehensive experimental findings, can be tested in future publications. Thus, while significant and newly incorporated into the revised manuscript, these new findings from the SDS-PAGE analysis do not alter the focus of our paper and we have not made any changes about the purpose and goals of the work.

As described above, we have made changes to the manuscript which are detailed below. To the end of the section on the proteomic analysis in the manuscript, we have added the following paragraph:

"The SDS-PAGE analysis showed that all three mucus present a prominent glycoprotein band (Supplementary Figures 9 – 11) at approximately 260 kDa, which was excised and subjected to proteomic analysis to identify the protein content (Supplementary Table 5). This analysis revealed that Collagen5, Collagen7, Novel4, a homolog of Novel4, which was named Novel4A, and two new proteins, deemed Novel7 and Novel8, which did not share similarity with any NCBI proteins, were found in these bands. Novel4, Novel7, and Novel8 were present in all samples. Further analysis of BLASTP results indicated Collagen5 and Collagen7 had similarity to oligomeric network-forming proteins.⁵⁸ PFAM domain search showed Collagen5 and Collagen7 contained von Willebrandt Factor A domains, calcium-binding regions, glycoprotein domains, disulfide-forming domains, and collagen domains.⁵⁹ Novel4A was found to be Ser- (19.9%), Thr- (11.1%), and Pro-rich (12.7%), and domain analysis of this protein revealed similarity to proline-rich extensin glycoprotein signatures, which are involved in crosslinking aromatic amino acids in plant cell walls. To further investigate the structures of these proteins, glycoproteomic site-mapping studies using tandem mass spectrometry on trypsin-digested peptides was employed to detect glycopeptides from the isolated mucus

protein samples and match these sequences to proteins isolated from the 260 kDa band (Supplementary Table 6). Novel4 released one glycopeptide modified with a HexNAcHex. Collagen5 presented six glycopeptides across all three mucus, three of which having two glycoforms, with the other three having one. Collagen5 also displays (Me)Hex₃HexNAc, (Me₂)Hex₂HexNAc, (Me)Hex₂HexNAc, Hex₂HexNAc, and HexHexNAc. Collagen7 was found to have one glycopeptide modified with (Me)Hex₂HexNAc. N-glycans were not detected on these proteins.”

Which refers to the following figures that we have also added to the Supporting Information:

Table S5. Proteomic quantification of proteins identified in the 260 kDa band of the SDS-PAGE gels shown in Figures S9 – 11. Percent abundance refers to percentage of each protein relative to total protein per sample within the band. Proteins of abundance greater than 5 % of total protein content are shown.

Protein	Accession	Mw	Percent Abundance		
			Adhesive	Lubricating	Protective
Collagen5	MM5_TRINITY_DN17478_c0_g1	65.0	0.0	0.0	5.9
Collagen7	MM5_TRINITY_DN19063_c0_g1	89.8	0.0	0.0	11.8
Novel4	MM4_TRINITY_DN21224_c1_g1	9.4	20.0	33.3	7.8
Novel4A	MM4_TRINITY_DN21224_c6_g1	46.3	0.0	11.1	7.8
Novel7	MM3_TRINITY_DN18065_c5_g2	31.1	40.0	33.3	9.8
Novel8	MM5_TRINITY_DN17583_c0_g1	46.6	20.0	11.1	2.0

Table S6. Glycoproteomic identification of proteins found in proteomic analysis of the 260 kDa band of the SDS-PAGE gels shown in Figures S9 – 11. Proteins from Table S5 not included here did not present detectable glycopeptides.

Protein	Peptide	Glycan	Sample
Novel4	P.KDQIS[+365.13220]DILK.K	HexNAc(1)Hex(1)	Protective
Collagen5	Q.IS[+689.23784][+14.01570]NKDVR.F	HexNAc(1)Hex(3) + Me	Protective
	G.LGFELQAIAS[+203.07937]NYK.N	HexNAc(1)	Adhesive
	A.IAS[+689.23784][+14.01570]NYK.N	HexNAc(1)Hex(3) + Me	Protective
	R.AGS[+527.18502]VINPK.E	HexNAc(1)Hex(2)	Protective
	R.AGSVINPKET[+527.18502][+28.03140]NK.C	HexNAc(1)Hex(2) + 2Me	Adhesive
	P.KET[+527.18502][+28.03140]NK.C	HexNAc(1)Hex(2) + 2Me	Lubricating
	R.GALFGLLFAT[+568.21157]EVQK.L	HexNAc(2)Hex(1)	Protective
	G.GLLFAT[+527.18502][+14.01570]EVQK.L	HexNAc(1)Hex(2) + Me	Adhesive
	P.PS[+365.13220]AKAAADDLK.S	HexNAc(1)Hex(1)	Protective
K.AAADDLKS[+527.18502][+28.03140]Q.G	HexNAc(1)Hex(2) + 2Me	Protective	
Collagen7	K.NT[+527.18502][+14.01570]FIK.A	HexNAc(1)Hex(2) + Me	Lubricating

We have also added the following paragraph to the methods section on **Glycoproteomic Tandem Mass Spectrometry** regarding analysis of the 260 kDa band:

“To identify glycoprotein candidates attributed to prominent 260 kDa bands in all three samples, peptides were extracted and analyzed by LC-MS/MS on the same instrument for glycoproteomic characterization. Entire duty cycle was used for stepped HCD fragmentation for high confidence peptide backbone sequencing. Common contaminants and decoys generated by Byonic software were added to the FASTA databases to interrogate candidate components in the bands. Glycoproteomics data were submitted to GlycoPost database under the accession number GPST000297.”

Finally, the author list was revised to include a new author, Xu Yang, who conducted these new experiments, and the Acknowledgements and Author Contributions sections have been updated accordingly.

Comment 5: It is also a concern that only 71 (only 64 reported in figure 2) out of +200-300 predicted proteins in Figure 2 was identified by proteomics. It raises the concern for how comprehensive the proteomic analysis is, even though the author claim it is an “inappropriate” comparison. I relate back to the authors own agenda that they want to do structure- function correlation, I think it is a valid point to ask if they have identified the components that are responsible for the macroscopic function of the different mucus samples.

Response: Here we believe that the Referee has misinterpreted the contents of Figure 2. Let us explain in the following two points how Figure 2 was generated and on the comparison that the Referee inaccurately describes as “inappropriate”.

- **Regarding the contents of Figure 2,** the “200-300 predicted proteins in Figure 2” that the Referee refers to are the proteins presented in the dendrogram for comparison purposes and are not ‘predicted’ as the Referee states. We identified 71 proteins in the isolated mucus, and all other entries in the dendrogram that make up the remaining “200-300” are previously reported proteins from the NCBI database that were used for comparison purposes. This is standard practice in the field: dendrograms, like the one shown in Figure 2, are made by conducting multiple sequence alignment between the genes of interest and a set of previously reported reference genes,¹⁹⁻²¹ and the purpose of this figure is to help classify the 71 proteins identified in the mucus. We explain all of this in detail in the methods section on “Bioinformatic Analysis” and in the “General Methods” section of the Supporting Information, including how we constructed the dendrogram in terms of which platforms we used, which procedures were employed, and which sequences were inputted, backed by a wealth of literature on using this approach for sequence alignments and analysis of genetic relationships.^{19,22-24} Although we believe our methods section, which we include below for reference, describes clearly how the dendrograms were constructed, we have edited the manuscript to increase clarity, and this change is described below.
- **Regarding what we had previously claimed was “inappropriate,”** the Referee judged the validity of our sequencing analysis by comparing the number of proteins identified in the proteomics to the number of genes identified from the transcriptomics. The proteomics data show *proteins found in the mucus* (71), while the transcriptomics show *all genes expressed by the animal cells* (179,552), which will naturally be a much larger number. Our finding of 71 proteins in the mucus is consistent with similar studies using integrated

transcriptomic-proteomic analysis on animal mucus. For example, a report on sea anemone mucus identified 62,880 transcripts from tentacle tissue, while the proteomics identified only 48 proteins in the mucus.²⁵ Another report on sea stars identified 97,945 transcripts and 244 mucus-specific proteins.²⁶ A third study on snail salivary glands found 115,171 transcripts and 119 proteins from the entire gland.²⁷ As our ratios of proteins found/genes expressed are consistent with the literature, we have made no changes to the manuscript regarding this point.

As noted above, we provide here the description in the manuscript of how the dendrogram was constructed. No changes have been made to this section.

“Bioinformatic Analysis” Methods:

“Using Clustal Omega within the EMBL–EBI web form (ebi.ac.uk/Tools/msa/clustalo/) and using default parameters, a multiple sequence alignment was conducted on our 71 proteins as well as an extensive set of reference proteins to generate a dendrogram and cluster the genes studied via neighbor-joining. For each protein type found, 3 – 5 proteins of the same type from other gastropods or mollusk species were selected from the NCBI non-redundant protein database and added to the alignment. Additionally, other protein types that appeared in the initial BLAST search results were included to build more accurate relationships.”

As described above, we have edited the section on “Identification and sequence alignment of snail mucus proteins” in the manuscript to address our first point above, such that we are completely explicit regarding the entries in the dendrogram. The sentence that was previously:

“To better understand the relationships between our sequenced proteins and those of other mollusks, and to determine the functions of the identified and unidentified proteins, global alignment analysis was employed to cluster the genes by sequence similarity (Figure 2, Supplementary Figure 8).”

Now reads:

“To better understand the relationships between our sequenced proteins and those of other mollusks, and to determine the functions of the identified and unidentified proteins, **dendrogram** analysis was employed to cluster the genes by sequence similarity **to each other and a set of NCBI reference proteins of similar families from mollusks** (Figure 2, Supplementary Figure 8).”

Comment 6: CAMPs: The authors claims that they have confirmed the size of the CAMPS by transcriptomics. The first question I would ask is how they know that the sequences they present are the fully translated sequence, none of them presents an N-terminus with an initiating Methionine. Glycosylation. Please provide spectra so that we clearly can see b/y ions as well as glycorelated fragment.

Response: Below, we address the Referee's comments regarding validation of CAMP translated sequences, CAMP initiating methionines, and CAMP glycosylation.

- **Regarding the validation of CAMP translated sequences**, it is first important to note that the sizes of the CAMPs were determined from transcriptomic data, proteomic data, glycoproteomic data, and, in response to comments from the previous set of Referees' comments, SDS-PAGE. These data are all consistent and all confirm the CAMP sequences with sufficient coverage. The computational package of Trimmomatic, Trinity, and SuperTranscripts assemble sequences into RNA transcripts, which will include open reading frames or, in other words, we are reporting predicted translated sequences, which is an accepted method of reporting transcriptomic data.²⁸⁻³² We provide further evidence for our predicted translated sequences using proteomics. We acknowledge that a limitation of *de novo* assembly, as generated by the program SuperTranscripts produces the largest unspliced isoform of each protein, so the true sizes may be shorter because of alternative splicing and post-translational cleavage. Additionally, it is known that *de novo* assemblers may be susceptible to redundancy and single-nucleotide errors, and an important area of research is refining these programs to improve sequencing data quality.^{33,34} Rectifying this limitation would require much deeper analysis into the RNA biology of these animals, which is outside the scope of this work. Verifying the size of the proteins via transcriptomics will always have this concern,^{35,36} which is why we conducted the SDS-PAGE-proteomic analysis to verify the sizes of the proteins directly, and these data were provided in the last set of responses. SDS-PAGE separates proteins from complex mixtures into bands of well-defined mass ranges, and the proteomic mass spectrometry that we provided alongside confirmed the contents of these bands.^{37,38} However, we agree with the Editor that it is important to note the limitations of this approach, and so we have revised the manuscript to note that the sizes predicted by the transcripts may differ from the actual protein.
- **Regarding the N-terminal methionines**, we disagree with the Referee's comment that "*none of them presents an N-terminus with an initiating Methionine*". We provide below the sequences of CAMP2 and CAMP3, with the full transcript sequence shown, and, as can be clearly seen from the sequence, the CAMPs do in fact present initiating methionines in the *N*-terminus region, which are shown in bold. Below, we also highlight sequences designating the exact peptides found for CAMP2 and CAMP3 in the proteomic analysis. These data have been present in the manuscript since it was first submitted. Additionally, the fact that "non-AUG start codons are used at an astonishing frequency" and translation initiation at alternative codon triplicates has been known since at least the 1980s³⁹⁻⁴² cannot be ignored. It is estimated that about 10% of sequences reported both in mammals and invertebrates (in RefSeq) can initiate translation at a non-AUG codon.⁴³ It is possible the true proportion of non-AUG-initiated proteins could indeed be much higher because the lack of an initiating AUG may cause many proteins to go unnoticed in genomic/transcriptomic analysis. Nevertheless, it is quite possible, and well-precedented, that some of our proteins may not have an initiating methionine. As this point serves to address only the Referee's concerns about our data and does not impact the findings, we have not made any changes regarding this point.
- **Regarding CAMP glycosylation**, we have edited Table S8 in the Supporting Information to demonstrate one example MS/MS spectrum of CAMP glycopeptides, with b/y ions labelled as well as the glycopeptide signal from the relevant fragment. The ten glycoproteomic MS/MS spectra have also been added to our data in GlycoPost.

CAMP Sequences:

CAMP2

hlshtdtqymcvtlgsmkedsisipihgcvfyqlqtdrtivglsvvc111slqqkvkesystqsqr1sl
qqkvkesssrqhqqpsprehqsfvfsatpitispgvtpelvtvrcGLEDDGNSGVSrvnsiiirtvdgsV
QKEVARIAYRQAATGGFSTEGASVTGDLSNKAGYLQITWPSPRHGLAGQYNCDIAALATVGDIVKfkssi
rvvstgkiadlslspafwsaqvkmmavqtrsnanaqktlrhrkrlgtvkgnlilrkrmirrisaavqftt
kralfraaililgenkkirDSDSWGEPLKARetvvevrfstsyarnpllvvsvvldadnttpgtrwswf
lpwfwmwliqktsdtelessdteleskiiryririrDVTPTGFKvvcgtwwdtilyridvrwvsnvarcl
shrhthvqs

CAMP3

11ldtllasiyqnmnrgficgvllvvsvttisqgerFVFSATPITISPGVTPELTVRCGLEDDGNSGVS
RVNSIIIRTVDGSVQKEVARIay

As stated above, we have made changes to the manuscript text to include limitations on the *de novo* assembly pipeline. The paragraph on the protein identification previously stated:

“To further validate the molecular masses of the identified proteins, SDS-PAGE was conducted on each of the three mucus to resolve polypeptides by size.”

Has been changed to:

“An important limitation of the *de novo* transcriptomic approach is that the program SuperTranscripts produces the largest unspliced isoform of each protein sequence and the expressed proteins may have a lower molecular mass than predicted.³¹ Therefore, to further validate the molecular masses of the identified proteins, SDS-PAGE was conducted on each of the three mucus to resolve polypeptides by size.”

Citing the reference:

31. Davidson, N. M., Hawkins, A. D. & Oshlack, A. SuperTranscripts: a data driven reference for analysis and visualisation of transcriptomes. *Genome biology* **18**, 1-10 (2017).

Also, as mentioned above we have edited **Table S8** to include a representative CAMP glycoproteomic MS/MS spectrum. Table S8, which was previously:

Table S6. Glycopeptidomic analysis of CAMPs.

Protein	Peptide	Glycan	Position
CAMP1	R.ALFRS[+673.24293]E.I	HexNAc(1)Hex(2)Fuc(1)	221
	C.GVFYLQT[+527.18502][+14.01570]DR.T	HexNAc(1)Hex(2) 14.0157	29
	R.ET[+673.24293]VVEVR.F	HexNAc(1)Hex(2)Fuc(1)	312
CAMP2	A.T[+406.15875]VGDIVK.F	HexNAc(2)	199
	L.QT[+203.07937]DR.T	HexNAc(1)	34
	V.LGS[+203.07937]M[+15.99492]K.E	HexNAc(1)	13
CAMP3	K.AVRDS[+527.18502]QGK.F	HexNAc(1)Hex(2)	74
	R.DS[+527.18502][+28.03140]QGK.F	HexNAc(1)Hex(2) 28.0314	77

Is now:

Table S8. Glycoproteomic analysis of CAMPs. Example tandem MS/MS spectrum of CAMP2, glycopeptide 3 is shown with b ions, y ions, and glycorelated fragment labelled.

Protein	Peptide	Glycan
CAMP1	R.ALFRS[+673.24293]E.I	HexNAc(1)Hex(2)Fuc(1)
	C.GVFYLT[+527.18502][+14.01570]DR.T*	HexNAc(1)Hex(2) + Me
CAMP2	R.ET[+673.24293]VVEVR.F	HexNAc(1)Hex(2)Fuc(1)
	A.T[+406.15875]VGDIVK.F	HexNAc(2)
	L.QT[+203.07937]DR.T	HexNAc(1)
	V.LGS[+203.07937]M[+15.99492]K.E**	HexNAc(1)
CAMP3	K.AVRDS[+527.18502]QGK.F	HexNAc(1)Hex(2)
	R.DS[+527.18502][+28.03140]QGK.F*	HexNAc(1)Hex(2) + 2Me

*Shift of S/T[+14.01570] and [28.03140] indicates methylated and demethylated glycans, respectively, O-linked to serine or threonine.

**Shift of M[+15.99492] indicates methionine oxidation to methionine sulfoxide

Comment 7: Please explain in the table S6 that +14 and +28 probably is methyl and +16 is oxidation of methionine as you did in the glycomic tables.

Response: We thank the Referee for their attention to detail. We have edited Table S6 (which is now Table S8) to show the glycan modifications indicated by the mass shift. Please see revised Table S6 (now Table S8) that are provided above in the response to Comment 6.

Comment 8: Did you only detect one glycoform of each peptide? Was the identification of these as stringent as the other proteomic data, or where there different criteria for their identification?

Response: The Referee brings up important questions regarding the results and technical details regarding CAMP glycoproteomic analysis, which we address in the following two points:

- **Regarding the CAMP glycoforms**, we detected that CAMP1 presented one glycopeptide with one glycoform, CAMP2 presented one glycopeptide with two glycoforms and three peptides with one glycoform, and CAMP3 presented one glycopeptide with two glycoforms, that met our criteria for selection. We agree with the Referee that these are important details about our CAMP analysis, and we have edited our section on CAMPs to include them in the manuscript, and these changes are listed below.
- **Regarding the criteria for glycopeptide identification**, the glycopeptides shown met equivalently stringent criteria, following similar standards set by the field,^{17,44} though glycoproteomics lacks universal standards for data collection and analysis compared to other -omics. As we had already stated our identification criteria in the Glycoproteomic Tandem Mass Spectrometry methods section of the manuscript, with the relevant quote included below for reference, we have not made any changes regarding this point.

“Glycoproteomic Tandem Mass Spectrometry:

A Byonic *N*-glycan database of insect and plant glycans as well as a tailored glycan list developed based on the glycomics results were searched as variable modification for *N*-glycans. A Byonic *O*-glycan database of 9 common *O*-glycans as well as a tailored glycan list developed based on our glycomics results were searched as variable modification for *O*-glycans. The data was then filtered based on a |log prob| value equal to or greater than 3, and a Delta Mod Score equal to or greater than 50.¹⁰¹ Matches were manually verified by confirming presence of oxonium ions and expected neutral loss patterns.”

As stated above, we have edited the manuscript's section on CAMPs to state our findings on CAMP glycoforms more explicitly. What was previously:

“Glycan site-mapping studies using tandem mass spectrometry on trypsin-digested CAMP peptides revealed that CAMPs are modified with several *O*-glycans (Supplementary Table 6). CAMP1 contained a FucHex₂HexNAc modification. CAMP2 was found to have several *O*-

glycans, including FucHex₂HexNAc, (Me)Hex₂HexNAc, HexNAc₂, and HexNAc. CAMP3 is modified with (Me₂)Hex₂HexNAc and Hex₂HexNAc.”

Now reads:

“**Glycoproteomic** site-mapping studies using tandem mass spectrometry on trypsin-digested CAMP peptides revealed that CAMPs are modified with several O-glycans (Supplementary Table 8). CAMP1 presented one glycopeptide containing a FucHex₂HexNAc modification. CAMP2 presented four glycopeptides, one with two glycoforms modified with (Me)Hex₂HexNAc and HexNAc, while the other three peptides contained FucHex₂HexNAc, HexNAc₂, and HexNAc. One glycopeptide was detected on CAMP3 with two glycoforms, modified with (Me₂)Hex₂HexNAc and Hex₂HexNAc, respectively.”

This section cites the following publication:

101. Bern, M., Kil, Y. J. & Becker, C. Byonic: Advanced Peptide and Protein Identification Software. *Current Protocols in Bioinformatics* **40**, 13.20.11-13.20.14, doi:[10.1002/0471250953.bi1320s40](https://doi.org/10.1002/0471250953.bi1320s40) (2012).

Comment 9: Figure 3B, how does the glycosylation reported there correspond to the glycosylation detected in table S6. Was the N-linked glycosylation also confirmed?

Response: We address the Referee’s concerns on CAMP glycosylation in the following two points on our glycomics findings and CAMP N-glycans.

- **Regarding the contents of Figure 3 and Table S6 (now Table S8)**, the O-glycans found on CAMP glycopeptides (Table S8) are consistent with the prevalent O-glycans found in the glycomic analysis (Figure 3a). The O-glycans, FucHex₂HexNAc, (Me₂)Hex₂HexNAc, (Me)Hex₂HexNAc, and Hex₂HexNAc, were detected on CAMP glycopeptides in the new *glycoproteomics* experiments and were all found previously in the *glycomic* analysis. As this information is clear when comparing **Figure 3** to **Table S6**, we have made no changes regarding this point.
- **Regarding CAMP N-glycans**, we did not detect N-glycosylation of CAMP peptides that met the criteria stated in *Comment 7*. We have edited the section on CAMPs to state explicitly this result.

As stated in our second point, we have added a sentence to the paragraph on CAMPs to the manuscript text in order to explicitly note that we did not detect N-glycans on CAMPs. The sentences:

“CAMP1 contained a FucHex₂HexNAc modification. CAMP2 was found to have several O-glycans, including FucHex₂HexNAc, (Me)Hex₂HexNAc, HexNAc₂, and HexNAc. CAMP3 is modified with (Me₂)Hex₂HexNAc and Hex₂HexNAc.”

Now read:

“CAMP1 contained a FucHex₂HexNAc modification. CAMP2 was found to have several O-glycans, including FucHex₂HexNAc, (Me)Hex₂HexNAc, HexNAc₂, and HexNAc. CAMP3 is modified with (Me₂)Hex₂HexNAc and Hex₂HexNAc. **No N-glycans were detected on CAMPs that met the criteria for identification.**”

Comment 10: In table S6 you had 1 O-linked for CAMP1, 5 for CAMP 2 and 2 for CAMP 3. How does this relate to the amount of O-linked and N-linked in figure 3B, or are these figures not relating to the actual analytical data?

Response: We thank the Referee for pointing out this detail on CAMP glycosylation in Figure 3b, which was not updated to reflect the new data. As the glycoproteomic analysis did not detect N-glycans on CAMPs, we have edited Figure 3b so that only O-glycosylation is represented.

Figure 3, which was previously:

Figure 3. Sequence analysis of CAMPs reveals similarity among N-terminal domains with interchangeable C-terminal functional domains. a) Multiple sequence alignment between I) *C. aspserum* mucus VWA protein, II) CAMP1 (truncated at C-terminus), III) CAMP2, IV) CAMP3. b) Schematic of CAMP architectures, showing conserved N-termini but varied C-termini between proteins.

Is now presented as:

Figure 3. Sequence analysis of CAMPs reveals similarity among *N*-terminal domains with interchangeable *C*-terminal functional domains. a) Multiple sequence alignment between I) *C. aspersum* mucus VWA protein, II) CAMP1 (truncated at *C*-terminus), III) CAMP2, IV) CAMP3. b) Schematic of CAMP architectures, showing conserved *N*-termini but varied *C*-termini between proteins.

Comment 11: Also, I don't agree that the proteomic confirms the size of the CAMPS, since mucins are often found to be proteolytically cleaved and smaller fragment can often be detected.

Response: We address the Referee's concerns on the sizes of CAMP proteins by making the following two points on proteolytic cleavage and on the SDS-PAGE-proteomics analysis.

- **Regarding mucin proteolytic cleavage**, we acknowledge that proteomics only detects peptide fragments, and not the full-length protein, meaning cleaved portions of proteins may not be detected. This challenge is standard to the field of shotgun proteomics and proteolytic cleavage is difficult to detect using this technique.^{46,47} Although researchers in the field are all familiar with this challenge, because this is an interdisciplinary manuscript and because it is important to describe limitations of our mucomics approach, we have added a statement to the manuscript stating this limitation.
- **Regarding the SDS-PAGE-proteomic analysis**, it is important to note again that we successfully validated the sizes of our proteins using the SDS-PAGE-proteomics analysis *that was suggested by the Referee* in the previous set of comments. As such, we have not made any changes regarding this point.

As stated above, we have edited the section "Identification and sequence alignment of snail mucus proteins" to more clearly state limitations of the approach. The sentences which read:

“Shotgun proteomic sequencing supported by a *de novo* assembled transcriptome identified proteins in the purified mucus samples. As *C. aspersum*’s genome has not yet been sequenced, a transcriptomic reference database of actively translated genes found in mucus-producing tissue was produced from RNA extracted from the foot and back tissue of whole *C. aspersum* snails.”

Now read:

“Shotgun proteomic sequencing supported by a *de novo* assembled transcriptome identified proteins in the purified mucus samples. As *C. aspersum*’s genome has not yet been sequenced, a transcriptomic reference database of actively translated genes found in mucus-producing tissue was produced from RNA extracted from the foot and back tissue of whole *C. aspersum* snails. It is important to note that shotgun proteomics only detects fragmented peptides and not the whole protein, and does not identify fragments that are present as a result of proteolytic cleavage.²⁹”

Citing the publication:

29. Rogers, L. D. & Overall, C. M. Proteolytic post-translational modification of proteins: proteomic tools and methodology. *Mol Cell Proteomics* **12**, 3532-3542, doi:10.1074/mcp.M113.031310 (2013).

Comment 12: Glycomics: The difference in lubrication/adhesiveness/protection and the length of the glycans is contradictory, since in the literature referred to in this subject actually shows that the glycans shown for the adhesive mucus is identical to the glycans they are referring to that provide the best lubricating molecules known in biological systems.

Response: While we disagree that the data is contradictory, we have made changes to the manuscript to clarify our comments regarding glycosylation. We would first note that we have already edited this section in response to previous reviews by the Referees, who were satisfied with the changes. See response to *Referee 2, Comment 17*, November 2022 revisions. We had edited the section to more accurately describe the findings in the literature on the role of Gal-rich glycans and glycan length in lubricity. No comments on this topic were made in the review of the November 2022 revisions, and this section has not been altered since then. There, we noted that our findings were consistent with the literature, provided extensive literature support, and noted that while some glycans are shared by adhesive and lubricating glycans, the adhesive glycans present more charged glycans, while the lubricating present more terminal galactose. Again, this was all described in detail previously, and no objection was made to that response previously.

We would hope that the Referee recognizes, as shown again here and throughout the responses to the Referees’ comments, that special care has been taken to ensure that our language appropriately reflects the previous data and literature as accurately as possible, and the great pains we have taken to avoid any unfounded conjectures. As stated above, we have edited the manuscript to ensure our claims are well-supported and do not overextend our results.

Nevertheless, to address the Referee's new concerns, in the second paragraph of the section, "Glycomic analysis of *C. aspersum* mucus", the sentences:

"Gal-rich glycans have a recognized role in biological lubricity, and increases in polysaccharide length are accompanied with increased material stiffness. Therefore, longer oligosaccharides likely contribute to its lubricative properties, while the shorter O-glycans found in the adhesive and protective mucus would attenuate lubrication."

Has been changed to:

"Gal-rich glycans have a recognized role in biological lubricity,⁶³ and increases in polysaccharide length are accompanied with increased material stiffness.⁶⁴ Therefore, longer, galactose-presenting oligosaccharides have been observed in biological lubricants and may increase the stiffness of these secretions."

Comment 13: N-Glycolyl neuraminic acid not very well substantiated (Figure S24), all samples appear to have Neu5Gc looking at the extracted ion chromatogram in S24. No MS/MS confirmation on glycomic or glycoproteomic level. Only evidence is the low intense mass found by glycomic analysis that we are supposed to believe corresponds to a composition containing Neu5Gc. It is also recommended that the authors stick to the minimum reporting standards for glycomics (MIRAGE) and it is advisable that they submit their data into repositories such as glycopost.

Response: We disagree that our claims regarding Neu5Gc are not well substantiated and would like to explain through the following points on the Neu5Gc findings we present, the substantiation of our claims, and the deposition of our raw data.

- **Regarding the manner with which we present our Neu5Gc findings**, we already had stated explicitly in the manuscript, "While low-abundance oxonium ions for Neu5Gc were detected in the tandem mass spectra, MS/MS analysis could not detect Neu5Gc-modified glycopeptides." This statement accurately reflects our findings, which is that Neu5Gc is found in low abundance compared to Neu5Ac and was found in one of the two analytical methods we used, and as such, this statement is an accurate reflection of our glycomics data. To provide a more detailed response: Table S8 details the glycomic MS results, and shows that Neu5Gc was found in the adhesive mucus only. Additionally, comparing the LC-MS/MS traces for HexNAc and Neu5Ac (Figures S22 and S23) to Neu5Gc (Figure S24), shows that Neu5Gc was identified, but with much lower signal intensity. Tables S11 – S13 show all of the glycans found on glycopeptides as determined by the glycoproteomic MS/MS analysis that met the identification criteria stated in the manuscript (as described in the response to *Referee 2, Comment 8*), which did not include Neu5Gc. All of these data are provided in the supporting materials. In summary, we already discuss in the manuscript the nuances of our data. We have data to support all the claims we are making and are as straightforward and open as possible about our confidence regarding the data. As such, we have not made any changes related to this point.

- **Regarding the substantiation of our claims**, tandem mass spectroscopy is a very sensitive and accurate analytical method, and the data is statistically significant, lending very strong support to our conclusions that Neu5Gc was present. Nevertheless, we still report that it was only detected by one of the two methods that we used to identify glycans. This further illustrates the great lengths we have already taken to provide the most accurate claims about our analysis of the mucus samples. As such, no changes were made in response to this comment because changes were already made to address this concern in the last round of reviews.
- **Regarding the deposition of our glycoproteomic data and MIRAGE standards**, we have deposited the new glycoproteomics data into GlycoPost under our existing posting (see response to comment 6). Thus, the glycomic and glycoproteomic data we report in our manuscript is available under a single accession number and is fully accessible by our readership. Additionally, our reporting of both the glycomic and glycoproteomic datasets were previously compliant with MIRAGE guidelines, indicating we had already met the stringent analytical standards needed to report our findings with confidence. We have edited the manuscript to indicate our glycoproteomic data are also accessible under the same deposition as our glycomic data.

As stated above, we have included the following statement in the manuscript and Supporting Information methods sections regarding the glycoproteomic tandem mass spectrometry:

“Glycoproteomics data were submitted to GlycoPost database under the accession number GPST000297.”

The data availability statement was also edited to reflect the data deposition, now stating:

“Glycomics data and glycoproteomics data were submitted to GlycoPost database under the accession number GPST000297.”

***Comment 14: Loading:** It is obvious that the samples subjected to SDS-PAGE and proteomic analysis is different, where the protective samples contained more material (Supplementary Figure 9-11) This is despite that the authors says that they loaded equal amount for the SDS-PAGE. This makes figures S12-14 difficult to compare the The authors did not describe in the experimental section if the shot gun proteomics suffered from the same problem, and if this caused a bias when comparing the proteins in the samples. The same for the glycomic analysis (here strangely they refer to the samples as “mucin” samples).*

Response: We would like to address the Referee’s concerns on the mucus protein analysis by making the following three points on the details of the SDS-PAGE experiments, the significance of our results, and on the contents of the -omics analyses.

- **Regarding the SDS-PAGE experiments**, we disagree that the samples used for the SDS-PAGE and gel band proteomics were loaded differently based on the gel images. All

samples were prepared for gel analysis identically by solubilizing them in ultrapure water, checking their concentration via Nanodrop spectroscopy, then diluting each sample in ultrapure water to a final concentration of 20 mg/mL. An equal volume of 2X SDS loading buffer was mixed into the sample, and Tween was added to a concentration of 1%. The samples were reduced at 95 C for 15 minutes and then loaded 10 uL onto the gel. At the same time, it is important to recognize that the gels are all running heterogenous samples derived from animals. Anyone knowledgeable in the art recognizes that it is nearly impossible for gels from different animal samples to appear identical, even if loaded identically, especially when the components and compositions of the different lubricating, adhesive, and protective samples vary so much. As such, we would be more concerned if the gels looked identical. Nevertheless, we have added details to the general methods of the Supporting Information to further clarify how the gels were prepared.

- **Regarding the significance of our SDS-PAGE findings**, it is important to note in response that SDS-PAGE analysis for mucus samples is notoriously challenging, *which is recognized by the Editor, Referees, and the literature*. It has been stated previously in the literature that mucus “cannot be analyzed by standard gel electrophoresis using conventional SDS-PAGE polyacrylamide gel and standard protein ladders”.⁴⁸⁻⁵⁰ Only as recent as 2021 has anyone reported SDS-PAGE of mucins¹⁸, which was on porcine gastric mucin, the best-characterized mucus model. Even this study, pioneered by Crouzier and Lieleg, who are key figures in mucin research, could only present appreciable signal after extensive purification and modification of mucins. So rather than noting minor differences in intensities of the stains, the Referee should recognize the triumph in overcoming one of the most vexing and unresolved challenges in mucus research these data represent. Here, we have successfully optimized conditions for SDS-PAGE analysis of mucus gels, such that the bands were clear enough that the Referee could pick them out from a photograph and clearly identify prominent bands, for example the band at ~260 kDa. In summary, these data show that we have solved a 50-year challenge in the field, now rendering one of the most fundamental techniques in biochemistry – SDS-PAGE – available for the analysis of one of the most prevalent biomaterials – mucus. As such, we have made no changes to the manuscript regarding this point.
- **Regarding the Referee’s concerns with the proteomic and glycomic analyses**, we did not encounter any issues in these experiments, and, as described in the first point, there was no evidence of bias in the experiments. Additionally, the glycomic analysis was conducted in a separate experiment, and thus the SDS-PAGE data (which began during the review process) could not cause bias in the glycan analysis since the glycomic data was included in the initial manuscript submission. As such, we have made no changes to the manuscript in response to this point. However, we do thank the Referee for pointing out our use of “mucin” in the methods sections, which we have edited to more accurately describe the samples tested.

As stated earlier, we have added the following sentences to the general methods of the Supporting Information on SDS-PAGE preparation:

Lyophilized snail mucus protein samples were suspended in 25 µL of ultrapure water and concentrations were verified via Nanodrop spectroscopy, and then diluted with ultrapure water to a concentration of 20 mg/mL. Samples were then mixed with an equal volume of 2X SDS loading buffer (Quality Biological, 351-082-661) with 5% (v/v) 2-mercaptoethanol (VWR, M31) and 1 % (v/v) Tween added.

Additionally, in the methods sections of the manuscript and supporting information, we have replaced uses of “mucin” with either “mucus” or “mucus proteins” as needed. For example, the sentences:

“Glycoproteomic Tandem Mass Spectrometry. Purified mucin samples were resuspended in 100 μ L of 50 mM ammonium bicarbonate.”

Now read:

“Glycoproteomic Tandem Mass Spectrometry. Purified **mucus protein** samples were resuspended in 100 μ L of 50 mM ammonium bicarbonate.”

Comment 15: Mucomics: I still don't see the necessity in introducing a -omic, for analysing mucus by various omics method. This sounds more what would be defined as multiomics of mucus or mucus multiomics. Otherwise we would have to define a new omics whenever we are analysing a new type of tissue or cell with a multiomics approach.

Response: We respectfully disagree that there is not a need to define “mucomics” here because, as stated in previous responses and in the manuscript, characterizing mucus *requires* the integration of protein sequencing, glycan identification, determining ion content, and measuring mechanical properties. These studies, taken together, generate comparable datasets that can be used to extract relationships between mucus structures and properties, which is exactly the purpose of all ‘omics’ analyses. The fact that so little is known about the structures and properties of such a ubiquitous biomaterial speaks to the need for the new -omics type approach that we describe in this manuscript. As such, we have not made any changes to the manuscript regarding this comment and stand fully behind our “mucomics” approach.

References

- 1 Snail Beauty Products Market - Market Trends, Analysis, and Forecast. *Prophecy Market Insights*, https://www.prophecymarketinsights.com/market_insight/Global-Snail-Beauty-Products-Market-4326 (2022).
- 2 Ahmed, I. A., Mikail, M. A., Zamakshshari, N. & Abdullah, A.-S. H. Natural anti-aging skincare: role and potential. *Biogerontology* **21**, 293-310 (2020).
- 3 Wang, J. V., Schoenberg, E., Saedi, N. & Ibrahim, O. Platelet-rich plasma, collagen peptides, and stem cells for cutaneous rejuvenation. *The Journal of clinical and aesthetic dermatology* **13**, 44 (2020).
- 4 Choi, F. D., Sung, C. T., Juhasz, M. & Mesinkovsk, N. A. Oral collagen supplementation: a systematic review of dermatological applications. *Journal of drugs in dermatology: JDD* **18**, 9-16 (2019).
- 5 Khalaji, S., Golshan Ebrahimi, N. & Hosseinkhani, H. Enhancement of biocompatibility of PVA/HTCC blend polymer with collagen for skin care application. *International Journal of Polymeric Materials and Polymeric Biomaterials* **70**, 459-468 (2021).
- 6 Li, G., Fukunaga, S., Takenouchi, K. & Nakamura, F. Comparative study of the physiological properties of collagen, gelatin and collagen hydrolysate as cosmetic materials. *International journal of cosmetic science* **27**, 101-106 (2005).
- 7 Rodrigues, C. V. *et al.* Potential of Atlantic Codfish (*Gadus morhua*) Skin collagen for skincare biomaterials. *Molecules* **28**, 3394 (2023).
- 8 Duffy, C. V., David, L. & Cruzier, T. Covalently-crosslinked mucin biopolymer hydrogels for sustained drug delivery. *Acta Biomaterialia* **20**, 51-59, doi:<https://doi.org/10.1016/j.actbio.2015.03.024> (2015).
- 9 Mane, P. C. *et al.* Terrestrial snail-mucus mediated green synthesis of silver nanoparticles and in vitro investigations on their antimicrobial and anticancer activities. *Scientific reports* **11**, 13068 (2021).
- 10 Di Filippo, M. F. *et al.* Cellulose derivatives-snail slime films: New disposable eco-friendly materials for food packaging. *Food Hydrocolloids* **111**, 106247 (2021).
- 11 Mencucci, R. *et al.* GlicoPro, Novel Standardized and Sterile Snail Mucus Extract for Multi-Modulative Ocular Formulations: New Perspective in Dry Eye Disease Management. *Pharmaceutics* **13**, 2139 (2021).
- 12 Trapella, C. *et al.* HelixComplex snail mucus exhibits pro-survival, proliferative and pro-migration effects on mammalian fibroblasts. *Scientific reports* **8**, 17665 (2018).
- 13 Deng, T. *et al.* A natural biological adhesive from snail mucus for wound repair. *Nature Communications* **14**, 396 (2023).
- 14 Camacho-Pacheco, A. V. *et al.* Feeding behavior, shrinking, and the role of mucus in the cannonball jellyfish stomolophus sp. 2 in captivity. *Diversity* **14**, 103 (2022).
- 15 Cornick, S., Kumar, M., Moreau, F., Gaisano, H. & Chadee, K. VAMP8-mediated MUC2 mucin exocytosis from colonic goblet cells maintains innate intestinal homeostasis. *Nature communications* **10**, 4306 (2019).
- 16 Gould, J., Valdez, J. W. & Upton, R. Adhesive defence mucus secretions in the red triangle slug (*Triboniophorus graeffei*) can incapacitate adult frogs. *Ethology* **125**, 587-591 (2019).
- 17 Yao, Y. *et al.* Mucus sialylation determines intestinal host-commensal homeostasis. *Cell* **185**, 1172-1188. e1128 (2022).
- 18 Marczynski, M. *et al.* Structural alterations of mucins are associated with losses in functionality. *Biomacromolecules* **22**, 1600-1613 (2021).
- 19 Sievers, F. & Higgins, D. G. The clustal omega multiple alignment package. *Multiple sequence alignment: Methods and protocols*, 3-16 (2021).
- 20 Nelesen, S., Liu, K., Zhao, D., Linder, C. R. & Warnow, T. in *Biocomputing 2008* 25-36 (World Scientific, 2008).

- 21 Thompson, J. D., Gibson, T. J. & Higgins, D. G. Multiple sequence alignment using ClustalW and ClustalX. *Current protocols in bioinformatics*, 2.3. 1-2.3. 22 (2003).
- 22 Blackshields, G., Sievers, F., Shi, W., Wilm, A. & Higgins, D. G. Sequence embedding for fast construction of guide trees for multiple sequence alignment. *Algorithms for Molecular Biology* **5**, 1-11 (2010).
- 23 Boyce, K., Sievers, F. & Higgins, D. G. Simple chained guide trees give high-quality protein multiple sequence alignments. *Proceedings of the National Academy of Sciences* **111**, 10556-10561 (2014).
- 24 Zhang, Y., Zhang, Q., Zhou, J. & Zou, Q. A survey on the algorithm and development of multiple sequence alignment. *Briefings in Bioinformatics* **23** (2022).
- 25 Ramírez-Carreto, S. *et al.* Transcriptomic and proteomic analysis of the tentacles and mucus of *Anthopleura dowii* Verrill, 1869. *Marine drugs* **17**, 436 (2019).
- 26 Hennebert, E., Leroy, B., Wattiez, R. & Ladurner, P. An integrated transcriptomic and proteomic analysis of sea star epidermal secretions identifies proteins involved in defense and adhesion. *Journal of proteomics* **128**, 83-91 (2015).
- 27 Bose, U. *et al.* Multiomics analysis of the giant triton snail salivary gland, a crown-of-thorns starfish predator. *Scientific reports* **7**, 1-14 (2017).
- 28 Haas, B. J. *et al.* De novo transcript sequence reconstruction from RNA-seq using the Trinity platform for reference generation and analysis. *Nature protocols* **8**, 1494-1512 (2013).
- 29 Raghavan, V., Kraft, L., Mesny, F. & Rigerte, L. A simple guide to de novo transcriptome assembly and annotation. *Briefings in Bioinformatics* **23**, doi:10.1093/bib/bbab563 (2022).
- 30 Grabherr, M. G. *et al.* Trinity: reconstructing a full-length transcriptome without a genome from RNA-Seq data. *Nature biotechnology* **29**, 644 (2011).
- 31 Davidson, N. M., Hawkins, A. D. & Oshlack, A. SuperTranscripts: a data driven reference for analysis and visualisation of transcriptomes. *Genome biology* **18**, 1-10 (2017).
- 32 Sewe, S. O., Silva, G., Sicat, P., Seal, S. E. & Visendi, P. in *Plant Bioinformatics: Methods and Protocols* 211-232 (Springer, 2022).
- 33 Cabau, C. *et al.* Compacting and correcting Trinity and Oases RNA-Seq de novo assemblies. *PeerJ* **5**, e2988 (2017).
- 34 Freedman, A. H., Clamp, M. & Sackton, T. B. Error, noise and bias in de novo transcriptome assemblies. *Molecular ecology resources* **21**, 18-29 (2021).
- 35 Hegde, P. S., White, I. R. & Debouck, C. Interplay of transcriptomics and proteomics. *Current opinion in biotechnology* **14**, 647-651 (2003).
- 36 Kumar, S. *et al.* Genetic variants of mucins: unexplored conundrum. *Carcinogenesis* **38**, 671-679 (2017).
- 37 Cao, X. *et al.* Comparative proteomic profiling of unannotated microproteins and alternative proteins in human cell lines. *Journal of proteome research* **19**, 3418-3426 (2020).
- 38 Takemori, A., Kaulich, P. T., Cassidy, L., Takemori, N. & Tholey, A. Size-Based Proteome Fractionation through Polyacrylamide Gel Electrophoresis Combined with LC-FAIMS-MS for In-Depth Top-Down Proteomics. *Analytical Chemistry* **94**, 12815-12821 (2022).
- 39 Andreev, D. E. *et al.* Non-AUG translation initiation in mammals. *Genome Biology* **23**, 111, doi:10.1186/s13059-022-02674-2 (2022).
- 40 Peabody, D. S. Translation initiation at non-AUG triplets in mammalian cells. *Journal of Biological Chemistry* **264**, 5031-5035 (1989).
- 41 Cao, X. & Slavoff, S. A. Non-AUG start codons: Expanding and regulating the small and alternative ORFeome. *Exp Cell Res* **391**, 111973, doi:10.1016/j.yexcr.2020.111973 (2020).
- 42 Kearse, M. G. & Wilusz, J. E. Non-AUG translation: a new start for protein synthesis in eukaryotes. *Genes Dev* **31**, 1717-1731, doi:10.1101/gad.305250.117 (2017).

- 43 Tikole, S. & Sankararamakrishnan, R. A survey of mRNA sequences with a non-AUG start codon in RefSeq database. *Journal of Biomolecular Structure and Dynamics* **24**, 33-41 (2006).
- 44 Shajahan, A., Supekar, N. T., Gleinich, A. S. & Azadi, P. Deducing the N-and O-glycosylation profile of the spike protein of novel coronavirus SARS-CoV-2. *Glycobiology* **30**, 981-988 (2020).
- 45 Chernykh, A., Kawahara, R. & Thaysen-Andersen, M. Towards structure-focused glycoproteomics. *Biochemical Society Transactions* **49**, 161-186, doi:10.1042/bst20200222 (2021).
- 46 Giglione, C., Boularot, A. & Meinel, T. Protein N-terminal methionine excision. *Cellular and Molecular Life Sciences CMLS* **61**, 1455-1474 (2004).
- 47 Rogers, L. D. & Overall, C. M. Proteolytic post-translational modification of proteins: proteomic tools and methodology. *Mol Cell Proteomics* **12**, 3532-3542, doi:10.1074/mcp.M113.031310 (2013).
- 48 Ramsey, K. A., Rushton, Z. L. & Ehre, C. Mucin agarose gel electrophoresis: Western blotting for high-molecular-weight glycoproteins. *JoVE (Journal of Visualized Experiments)*, e54153 (2016).
- 49 Segrest, J., Jackson, R., Andrews, E. & Marchesi, V. Human erythrocyte membrane glycoprotein: a re-evaluation of the molecular weight as determined by SDS polyacrylamide gel electrophoresis. *Biochemical and biophysical research communications* **44**, 390-395 (1971).
- 50 Tytgat, K. M. *et al.* Unpredictable behaviour of mucins in SDS/polyacrylamide-gel electrophoresis. *Biochem J* **310 (Pt 3)**, 1053-1054, doi:10.1042/bj3101053 (1995).